# Molecular dynamics of the matrisome across sea anemone life history

Bruno Gideon Bergheim[1], Alison G Cole[2], Mandy Rettel[3], Frank Stein[3], Stefan Redl[4], Michael W Hess[5], Aissam Ikmi[3], Suat Özbek[1]*

[1]University of Heidelberg, Centre for Organismal Studies, Department of Evolutionary Neurobiology, Heidelberg, Germany; [2]Department of Neurosciences and Developmental Biology, Faculty of Life Sciences, University of Vienna, Vienna, Austria; [3]European Molecular Biology Laboratory, Heidelberg, Germany; [4]Institute of Neuroanatomy, Medical University of Innsbruck, Innsbruck, Austria; [5]Institute of Histology and Embryology, Medical University of Innsbruck, Innsbruck, Austria

## eLife Assessment

This **valuable** study provides a comprehensive description of the *Nematostella vectensis* matrisome - the genes encoding the proteins of the extracellular matrix. The authors combine new mass spectrometry data with bioinformatic analyses of previously published genomic and single-cell RNAseq data. The analysis is thorough, and the discussion and conclusions are **convincing**. This work will be of interest to biologists working on the evolution of the matrisome, as well as more broadly those working with non-bilaterian animals.

**\*For correspondence:**
suat.oezbek@cos.uni-heidelberg.de

**Competing interest:** The authors declare that no competing interests exist.

**Abstract** The evolutionary expansion of extracellular matrix (ECM) molecules has been crucial for the establishment of cell adhesion and the transition from unicellular to multicellular life. Members of the early diverging metazoan phylum Cnidaria offer an exceptionally rich perspective into the metazoan core adhesome and its original function in developmental and morphogenetic processes. Here, we present the ensemble of ECM proteins and associated factors for the starlet sea anemone *Nematostella vectensis* based on in silico prediction and quantitative proteomic analysis of decellularized mesoglea from different life stages. The integration of the matrisome with single-cell transcriptome atlases shows that gastrodermal cells are the primary producers of *Nematostella's* complex ECM, confirming the homology of the cnidarian inner cell layer with bilaterian mesoderm. The transition from larva to polyp is marked by an upregulation of metalloproteases and basement membrane components including all members of an unusually diversified SVEP1/Polydom family, suggesting massive epithelial remodeling. The enrichment of Wnt/PCP pathway factors during this process further indicates directed cell rearrangements as a key contributor to the polyp's morphogenesis. Mesoglea maturation in adult polyps involves wound response proteins indicating shared molecular patterns in growth and regeneration. Our study identifies conserved matrisomal networks that coordinate transitions in *Nematostella's* life history.

## Introduction

The evolution of extracellular matrix, a complex network of secreted, typically modular proteins, is closely linked to the emergence of metazoan life forms (*Hynes, 2009*; *Hynes, 2012*; *Naba, 2024*; *Ozbek et al., 2010*; *Rokas, 2008*). While some ECM components, such as integrins and cadherin receptors, can be traced back to unicellular organisms (*Nichols et al., 2012*; *Sebé-Pedrós et al., 2010*), early diverging cnidarians (hydroids, jellyfish, corals, and sea anemones) are believed to possess

one of the most complete adhesomes among non-bilaterian clades (*Ozbek et al., 2010*; *Tucker and Adams, 2014*). Cnidarians, the sister group to bilaterians, are characterized by a simple body plan with a central body cavity and a mouth opening surrounded by tentacles. They are diploblastic organisms, consisting of an outer epithelium and an inner gastrodermis separated by a complex ECM called mesoglea (*Bergheim and Özbek, 2019*; *Sarras, 2012*). The mesoglea, which is best studied in the freshwater polyp *Hydra*, forms a flexible, tri-laminar structure. It is composed of a central, amorphous interstitial matrix (IM) interspersed with collagenous fibrils, sandwiched between two thin layers of basement membrane (BM) (*Aufschnaiter et al., 2011*; *Bergheim and Özbek, 2019*; *Sarras et al., 1991*; *Shimizu et al., 2008*). Studies in *Hydra* have shown that the mesoglea can be separated intact from the epithelial cell sheets by a freeze-thaw technique (*Day and Lenhoff, 1981*; *Veschgini et al., 2023*). Previously, we analyzed the proteome of decellularized *Hydra* mesoglea and identified 37 unique protein sequences (*Lommel et al., 2018*), including most of the described core matrisome components (*Sarras, 2012*). Among medusozoans (jellyfish and hydroids), hydras stand out for having lost the free-swimming medusa form. They also lack a planula larva stage from which anthozoans (corals and sea anemones) typically produce sessile polyps. We therefore hypothesized that a cnidarian species with a complex life cycle could offer a more comprehensive picture of the non-bilaterian ECM repertoire. Here, we analyzed the matrisome of the anthozoan starlet sea anemone *Nematostella vectensis* by employing in silico predictions of ECM proteins that were partially confirmed by a subsequent proteomic analysis of decellularized mesoglea. We detected a rich collection of matrisome proteins, comparable in its core matrisome complexity to vertebrate species (*Naba et al., 2012*). Furthermore, mapping of our matrisome data onto a previously established single-cell transcriptome dataset (*Cole et al., 2024*; *Steger et al., 2022*) revealed a prominent role of the gastroderm in ECM production. Cnidocytes, which produce the cnidarian stinging organelle, are characterized by a distinct set of ECM proteins that significantly contribute to the complexity of the cnidarian matrisome. Quantitative proteomics of mesoglea samples from different life stages (larva, primary, and adult polyp) showed that, while the larval mesoglea contained only a few exclusively enriched factors, the transition from the larval stage to primary polyp was marked by an upregulation of a large fraction of the matrisome. This set of proteins included many metalloproteases and basement membrane factors, indicating significant epithelial reorganization. Remarkably, all members of an unusually diverse SVEP1/Polydom family were upregulated during this morphogenetic process, implicating a conserved role of this protein family for epithelial morphogenesis. Additionally, a significant enrichment of Wnt/planar cell polarity (PCP) signaling components, such as ROR2 and protocadherin Fat4, supports that directed cell movements underlie the axial elongation and morphogenesis of the polyp (*Stokkermans et al., 2022*). The final transition to the adult animal involves an increased addition of elastic fiber components to the mesoglea and matricellular factors associated with wound healing, indicating common ECM-associated mechanisms in regeneration, growth, and tissue differentiation.

## Results

### Molecular composition and structure of the *Nematostella* ECM

To investigate the components and dynamics of the ECM throughout various life stages of *Nematostella vectensis*, we employed a protocol for obtaining decellularized mesoglea, originally developed for *Hydra* (*Day and Lenhoff, 1981*; *Lommel et al., 2018*; *Veschgini et al., 2023*). We isolated mesogleas from larvae at three days post-fertilization (3 dpf), primary polyps at 10 dpf, and from small adult polyps that were at least 1 year old. Protein extraction was performed under strongly reducing conditions and at high temperatures (90°C) to solubilize the cross-linked protein network of the ECM. The extracted proteins were digested with trypsin and analyzed using quantitative mass spectrometry (*Figure 1A*). Extensive studies in both vertebrates and invertebrates have identified specific characteristics of ECM proteins, typically based on conserved domains and domain arrangements (*Engel, 1996*; *Hohenester and Engel, 2002*; *King et al., 2008*; *Naba et al., 2012*; *Ozbek et al., 2010*; *Tucker and Adams, 2014*). Building on this knowledge and a de novo annotation of all predicted *Nematostella* protein models using InterProScan (*Jones et al., 2014*), we identified 1812 potential ECM proteins (*Figure 1B*). These were further classified into orthogroups based on similarity using OrthoFinder (*Emms and Kelly, 2019*). To refine this analysis, we also predicted in silico matrisomes from protein models of several early-branching metazoan species, including two choanoflagellates, two sponges,

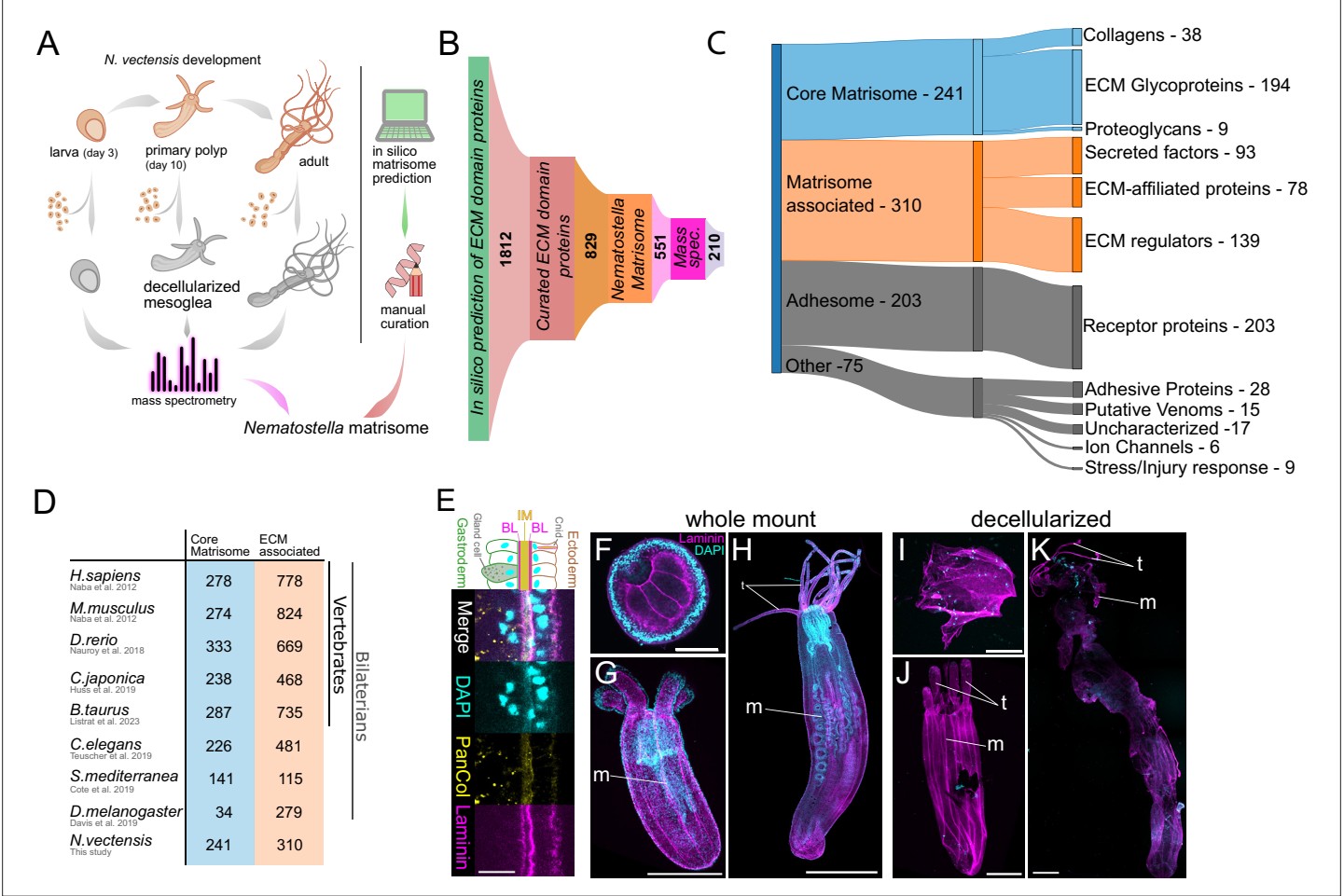

**Figure 1.** Analysis of the *Nematostella* matrisome. (**A**) Mesoglea from larvae, primary polyps, and adults was decellularized and analyzed by mass spectrometry. In parallel, an in silico matrisome was predicted using a computational approach and curated manually. (**B, C**) 1812 potential ECM proteins were predicted bioinformatically. The manually curated list of ECM factors consists of 829 proteins. The curated ECM proteins were sorted into core matrisome and matrisome-associated groups, which together constitute the *Nematostella* matrisome (551 proteins). The remaining non-matrisomal proteins are categorized as 'adhesome' that include transmembrane receptors, and 'other' ECM domain-containing proteins, which include adhesive proteins, venoms, enzymes, ion channels, stress and injury response factors, and diverse uncharacterized proteins (see ***Supplementary file 1*** for detailed annotations and sub-categories). In total, 287 ECM proteins were confirmed by mass spectrometry analysis, 210 of which belong to the matrisome and 47 to the 'adhesome'. (**D**) Comparison of the *Nematostella* matrisome size with published matrisomes of other species. While the complexity of the *Nematostella* core matrisome is comparable to that of vertebrates, the number of ECM-associated proteins is disproportionally lower. The *Drosophila* core matrisome is characterized by significant secondary reduction. (**E**) Laminin antibody stains the bilaminar structure of the BL (magenta) at the base of the epithelial cell layers, while the pan-Collagen antibody (yellow) detects the central IM. Scale bar, 10 μm. The three life stages of *Nematostella* before (**F–H**) and after (**I–K**) decellularization. The mesoglea is stained with Laminin antibody to demonstrate its structural preservation and by DAPI (cyan) to visualize residual nuclei and nematocysts. The decellularized mesoglea retains morphological structures such as tentacles (t) and mesenteries (m). Scale bars: **F**, **G**, **I**, **J**, 100 μm; **H**, **K**, 1 mm.

The online version of this article includes the following source data and figure supplement(s) for figure 1:

**Figure supplement 1.** Protein complexity in mesoglea samples of *Nematostella* and *Hydra* resolved by one-dimensional SDS-PAGE.

**Figure supplement 1—source data 1.** Uncropped gel images of *Hydra* (left) and *Nematostella* (right) mesoglea preparations.

**Figure supplement 1—source data 2.** Uncropped and labeled gel images of *Hydra* (left) and *Nematostella* (right) mesoglea preparations.

**Figure supplement 2.** Epitope sequences of laminin and collagen antibodies.

**Figure supplement 3.** Ultrastructure of the mesoglea in *Nematostella* larvae (**A–F**) and primary polyps (**G–P**).

three ctenophores, nine additional representative cnidarians, and two placozoan species. SignalP (*Teufel et al., 2022*) was used to confirm the presence of signal peptides, while DeepLoc (*Thumuluri et al., 2022*) was employed to predict the cellular localization of the matrisome draft. Additionally, we explored the closest BLAST hits in the NCBI and SwissProt databases. After manually excluding duplicates and sequencing artifacts, we narrowed down the number of high-confidence ECM genes defined by the possession of a *bona fide* ECM domain (*Hynes and Naba, 2012*) to 829 (*Figure 1B*). Each protein sequence was manually reviewed, with annotations assigned based on domain predictions, UniProt identifiers, and comparisons to known ECM proteins. Our subsequent mass spectrometry analysis initially identified a total of 5286 proteins. However, we suspected that a significant portion might be cellular contaminants originating from residual amoebocytes within the mesoglea (*Tucker et al., 2011*), which could not be completely removed during the isolation procedure. To address this, we used our curated in silico list of 829 candidate ECM genes as a filter and identified 287 ECM components in the isolated *Nematostella* mesogleas, of which 210 belonged to the matrisome (*Figure 1B*, *Supplementary files 1 and 2*). This number exceeds the previously reported 37 matrisomal proteins in *Hydra* (*Lommel et al., 2018*), suggesting a greater compositional diversity in *Nematostella*. A comparison of mesoglea samples of both species resolved by protein gel electrophoresis confirmed the higher complexity of the *Nematostella* mesoglea, particularly in the lower molecular weight fraction (*Figure 1—figure supplement 1*). The discrepancy between the number of predicted ECM proteins and those confirmed by mass spectrometry can be attributed to the exclusion of nematocysts from the mesoglea isolates, despite their substantial contribution to the matrisome (as shown below). Furthermore, soluble factors and transmembrane receptors are likely underrepresented in the mesoglea isolates. We further organized our curated dataset following the classification proposed by Hynes and Naba, who introduced the concept of a 'core' matrisome, characterized by 55 signature InterPro domains, including EGF, LamG, TSP1, vWFA, and collagen (*Hynes and Naba, 2012*). Within this core matrisome, we identified 241 proteins, including collagens, proteoglycans, and ECM glycoproteins (e.g. laminins; *Fahey and Degnan, 2012*) and thrombospondins (*Shoemark et al., 2019*; *Tucker et al., 2013*; *Figure 1C*, *Supplementary file 1*). Additionally, we identified a set of 310 'matrisome-associated' factors. This group includes molecules that (i) have structural or functional associations with the core matrisome, (ii) are involved in ECM remodeling (e.g. metalloproteases), or (iii) are secreted proteins, including growth factors. In summary, the *Nematostella* matrisome comprises 551 proteins, representing approximately 3% of its proteome (*Artamonova and Mushegian, 2013*) and roughly half the size of the human matrisome with 1056 proteins (*Naba et al., 2012*). While the *Nematostella* core matrisome is comparable to that of bilaterians, vertebrate species exhibit a dramatic expansion in ECM-associated factors (*Figure 1D*). The remaining non-matrisomal 278 proteins, identified through negative selection based on exclusive domain lists for each ECM category (*Naba et al., 2012*), were assigned to the 'adhesome' category primarily comprising transmembrane adhesion receptors (e.g. cadherins and IgCAM-like molecules) or categorized as 'other' ECM domain-containing proteins including proteins with specialized functions, such as venoms or stress and injury response proteins (*Figure 1B and C*, *Supplementary file 1*). Additionally, 17 proteins could not be confidently assigned to any category but were retained in the dataset as candidates for future functional characterization.

To validate the preservation of the isolated ECM, we generated polyclonal antibodies targeting unique peptide sequences in laminin gamma 1 (anti-Lam), type IV collagen NvCol4b (anti-Col4), a specific fibrillar collagen NvCol2c (anti-Col2), and a consensus motif for several *Nematostella* fibrillar collagens (anti-PanCol; *Figure 1—figure supplement 2*). We then performed immunofluorescence confocal microscopy on *Nematostella* whole mounts and decellularized mesoglea from all life stages (*Figure 1E–K*). Consistent with recent findings in *Hydra* (*Veschgini et al., 2023*), the in vivo and ex vivo images exhibited similar patterns along the polyp's body, indicating the structural integrity of the mesoglea after decellularization (*Figure 1F–K*). In cross-sections, the laminin antibody stained the thin double layer of the basement membrane (BM) lining the two epithelia, while the PanCol antibody detected the intervening fibrous layer of the interstitial matrix (IM; *Figure 1E*). Both antibodies also showed diffuse staining at the apical surface of ectodermal cells, likely due to the sticky nature of the glycocalyx. Ultrastructural examination of the mesoglea revealed a mean thickness of approximately 0.5 μm in larvae and 1.5 μm in primary polyps (*Figure 1—figure supplement 3A–B, G–H, Supplementary file 3*). In the triangular areas at the base of the gastrodermal mesenteric folds, the mesoglea

was expanded (*Figure 1—figure supplement 3C–F*). The BM lining the epithelia of larvae was very delicate, measuring only 70 nm, and the IM appeared as a loose array of thin fibrils (*Figure 1—figure supplement 3B*). Primary polyps possessed a distinct BM, with a thickness of approximately 130 nm, appearing as a dense meshwork of fibrils (*Figure 1—figure supplement 3H*, *Supplementary file 3*). Previous reports (*Tucker et al., 2011*) indicated that the IM in older primary polyps was interspersed with thin fibrils of about 5 nm and extended thick fibrils of about 20–25 nm. Our samples from younger primary polyps (*Figure 1—figure supplement 3H*) showed thin fibrils of approximately 6 nm and thick ones of about 13 nm, with occasional fibrils of up to 27 nm (*Supplementary file 3*). Immunostainings with the laminin antibody revealed a thickened mesoglea at the aboral pole of the polyp, forming an unusual knot-like structure (*Figure 1—figure supplement 3J*). Ultrastructural analysis showed that the fibrils in this region were densely packed and aligned along the oral-aboral axis (*Figure 1—figure supplement 3K–L*), possibly providing a rigid attachment site for the mesentery retractor muscles. Immunoelectron microscopy confirmed the observations from immunofluorescence: anti-Lam immunogold labeling was primarily localized along the plasma membrane of ECM-lining cells, anti-Col4 labeling was found at the BM (*Figure 1—figure supplement 3M–N*), and anti-PanCol labeling was predominantly distributed throughout the IM, with a similar pattern observed for the anti-Col2 antibody (*Figure 1—figure supplement 3O–P*).

## Cell-type specificity of matrisome expression

Recently, single-cell RNA sequencing has been applied to identify the origin of neuroglandular cell lineages in *Nematostella* (*Steger et al., 2022*), hypothesize on the origin of muscle cell types (*Cole et al., 2023*), and catalog the distribution of cell states associated within all tissue types (*Cole et al., 2024*). We made use of this developmental cell type atlas to determine the cell-type specificity of matrisomal gene expression. Expression profiles for all ECM genes across the entire life cycle are available in the supplementary data (*Supplementary file 4*). We calculated an average expression score for each of the ECM gene sets (core and associated matrisome, adhesome/other) and found above average scores for the core matrisome associated with the mature gastrodermis and developing cnidocytes, and to a lesser extent also for the other two categories (*Figure 2A and B*). Core matrisome genes also showed additional high expression scores within an uncharacterized gland cell type (GD.1), matrisome-associated genes within the digestive gland set, and adhesome/other genes within the maturing cnidocytes (*Figure 2A and B*). We looked specifically at the distribution of core matrisome genes across all cell states and generated a list of differentially up-regulated genes (*Supplementary file 5*; *Figure 2—figure supplement 1*). Of note is the absence of any differentially expressed core matrisome factors within the ectodermal tissues, and contrastingly, a large set that are specific to either the mesoendodermal inner cell layer (gastrodermis) or cnidocytes (*Figure 2B*). We also find sets of core matrisome genes that are specific to different secretory gland cell types, including mucin-producing, digestive-enzyme producing, and uncharacterized S2-class cell types. Interestingly, the gland cell-specific matrisome genes include several of the Polydom family members upregulated during larva-to-primary polyp transition (see below). Altogether, these expression profiles suggest that core components of the mesoglea (collagens, laminins) are produced from the inner cell layer, and that a large set of ECM glycoproteins and all of the minicollagens *David et al., 2008*; *Zenkert et al., 2011* have been recruited into the formation of the cnidarian synapomorphy, the cnidocytes.

Collagens constitute the primary structural components of the animal ECM and, being a highly diverse family of triple helical proteins (*Fidler et al., 2018*), form a significant part of the core matrisome. Vertebrate collagens consist of 28 types (I-XXVIII) categorized as fibril-forming, network or beaded filament-forming, and transmembrane collagens (*Kadler et al., 2007*). Type IV collagen, a major constituent of basement membranes, is considered to be a primordial component of the animal ECM based on studies in ctenophores that lack fibrillar collagens but display a remarkable diversity of collagen IV genes (*Draper et al., 2019*; *Fidler et al., 2017*). Our analysis revealed 12 bilaterian-type collagens (*Figure 2C*) with conserved C-terminal non-collagenous trimerization domains (NC1) and extended triple helical stretches of ~1000 residues, including two sequences for type IV collagen (NvCol4a/b) together with a peroxidasin homolog (NV2.13306), indicating the presence of sulfilimine cross-links that stabilize the *Nematostella* BM (*Fidler et al., 2014*). In *Hydra*, which lacks this specific post-translational modification, six collagen genes have been identified through cDNA cloning and proteomic analysis (*Deutzmann et al., 2000*; *Fowler et al., 2000*; *Lommel et al., 2018*; *Zhang et al.,*

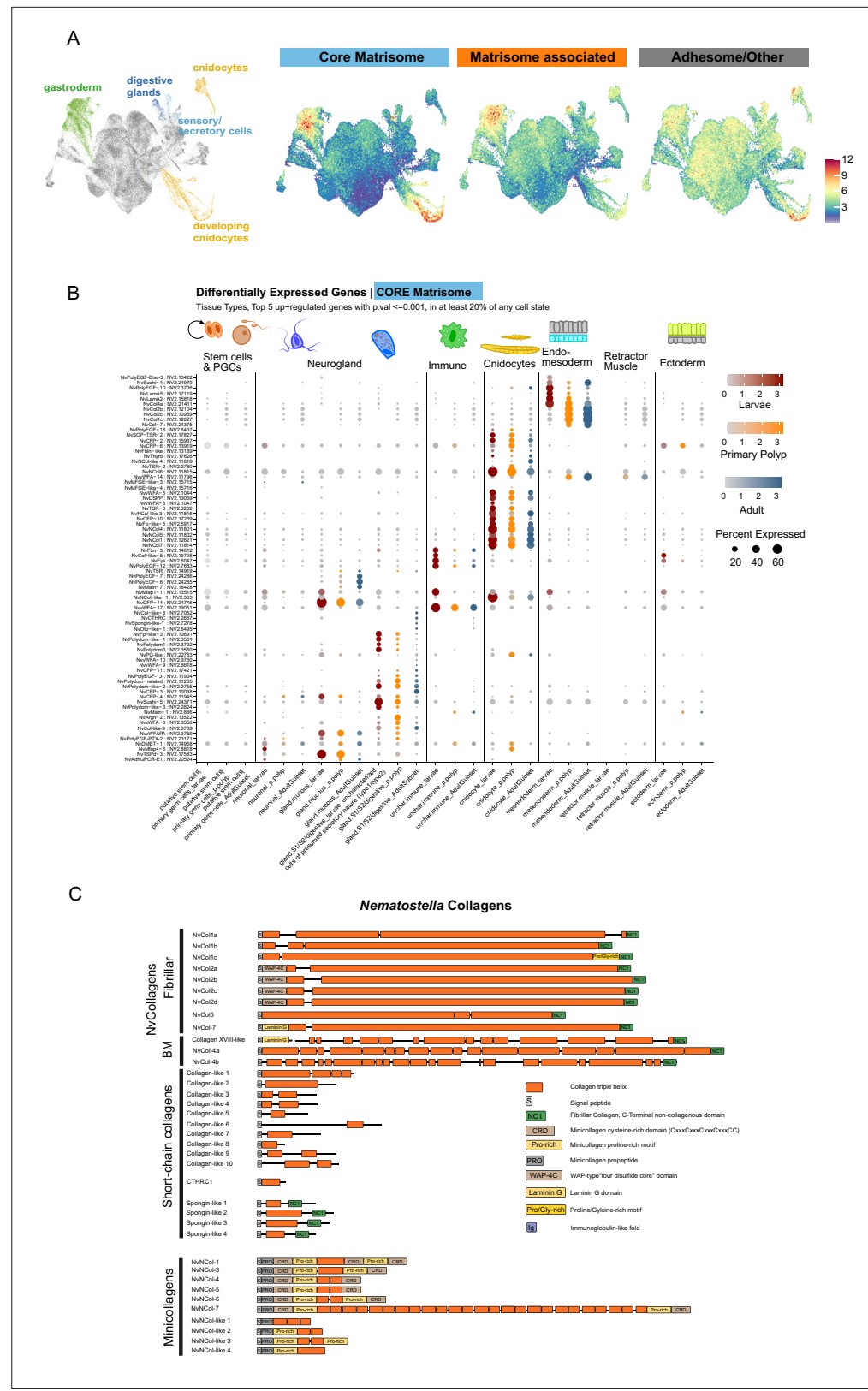

**Figure 2.** Single cell atlas of core matrisome genes. (**A**) Dimensional reduction cell plot (UMAP) highlighting cell clusters showing over-abundant expression of the core matrisome, matrisome-associated, and adhesome/other gene sets. Expression values correspond to gene module scores for each set of genes. (**B**) Dotplot expression profiles of upregulated genes of the core matrisome across cell type partitions, separated across phases of the

*Figure 2 continued on next page*

*Figure 2 continued*

life cycle. Illustrated are the top 5 genes with expression in at least 20% of any cell state cluster, calculated to be upregulated with a p-value of ≤ 0.001. See **Supplementary file 5** for a full list of differentially expressed core matrisome genes. Larva (red colour scale) = 18 hr:4 day samples; Primary Polyp (orange colour scale) = 5:16 day samples; Adult (blue colour scale) = tissue catalog from juvenile and adult specimens. (**C**) *Nematostella* collagens. Domain organization of matrisome proteins containing a collagen triple helix as core element. The proteins are categorized into fibrillar and basement membrane collagens, short-chain collagens, and nematocyte-specific minicollagens.

The online version of this article includes the following figure supplement(s) for figure 2:

**Figure supplement 1.** Expression profiles of core matrisome genes.

**Figure supplement 2.** Single-cell atlas of collagen genes.

*2007*). We have classified the collagens according to the *Hydra* nomenclature (*Zhang et al., 2007*) and identified three isoforms of NvCol1, each comprising an isolated minor triple helical domain at the N-terminus (*Figure 2C*, upper panel). In addition, we detected four NvCol2 paralogs, which contain an additional whey acidic protein 4 disulfide core (WAP) domain at the N-terminus. However, our dataset lacks a Hcol3-like collagen with alternating N-terminal WAP and vWFA domains, although it includes a protein with extensive WAP/vWFA repeats but lacking a collagen triple helix or NC1 domain (NV2.11346). NvCol5 consists of a continuous collagen domain with two minor interruptions following the signal peptide and is otherwise similar to NvCol1. We have not found sequences resembling Hcol6, which is characterized by triple helical sequences interrupted by multiple vWFA domains. Collagen XVIII-like, which is predicted to contain an unrelated Mucin-like insertion, includes an N-terminal Laminin-G/TSPN motif followed by a discontinuous central collagenous domain, similar to the BM-associated vertebrate α1(XVIII) collagen chains (*Heljasvaara et al., 2017*) and *Drosophila* Multiplexin (*Meyer and Moussian, 2009*). NvCol7 shares a similar domain organization but differs by having only a single interruption of the triple helix near the N-terminus. According to Exposito et al., the *Nematostella* fibrillar collagen sequences are phylogenetically related to A clade collagens that in mammalians possess a vWFC module in their N-propeptide supposed to have evolved from the cnidarian WAP domain (*Exposito et al., 2008*). NvCol7 is an exception and belongs to B clade collagens characterized by the possession of an N-terminal Laminin-G/TSPN domain.

Most of the *Nematostella* collagens show a broad expression in diverse gastrodermal cell populations throughout all life stages (*Figure 2—figure supplement 2*). NvCol2c is an exception, as it is predominantly expressed in neuronal cells of larvae, indicating a specialized function in neurogenesis. NvCol7 is distinguished by showing an expression both in gastrodermal cells and two small cnidocyte cell populations described by Steger et al. to be exclusive for planula larvae (*Steger et al., 2022*). Unlike the IM collagens, NvCol4a is additionally expressed during embryogenesis, emphasizing the pivotal role of the BM in organizing the epithelial tissue architecture during early development. In addition to these bilaterian-type collagens, we have identified four spongin-like proteins, which are short-chain collagens with derived NC1 domains (*Figure 2C*, middle panel). These molecules have been described as truncated variants of collagen IV with a wide distribution in several invertebrate phyla (*Aouacheria et al., 2006*). Except for NvSpongin-like-2, their expression is not aligned with that of fibrillar collagens, but is mostly restricted to neuroglandular cells (*Figure 2—figure supplement 2*). An additional set of 'collagen-like' sequences that comprise short triple helical stretches without additional domain motifs (*Figure 2C*, middle panel) is broadly expressed across cell types and developmental stages (*Figure 2—figure supplement 2*, *Supplementary file 4*). In contrast, the large set of minicollagens (*Figure 2C*, lower panel) is strictly aligned with the cnidocyte lineage as detailed below.

## A specialized cnidocyte matrisome

Cnidocytes, the stinging cells of cnidarians, are a key synapomorphy of this clade. Previous studies have demonstrated that, from a transcriptomic perspective, nematocyst capsule formation is distinct from the mature profile (*Chari et al., 2021*; *Steger et al., 2022*). We filtered the dataset of the 829 curated ECM protein-coding sequences for genes expressed within cnidocytes and found 298 genes that are detectable above average in at least 5% of any cnidocyte transcriptomic state (*Figure 3—figure supplement 1*, *Supplementary file 6*). We further separated this list of genes into those that are absent from non-cnidocyte states (101 genes: 'exclusive') or are also detected within more than

50% of the non-cnidocyte transcriptomic states (41 genes: 'ubiquitous'; *Figure 3A*). We consider the remaining genes shared across cnidocyte and non-cnidocyte profiles (156: 'shared'). Of the cnidocyte-specific genes, we examined the proportion of genes specific to either the capsule-building specification profiles (79: 'specification') or the maturation phase of cnidogenesis (24: 'maturation'). Most of these genes are restricted to the specification phase, with a smaller subset associated with the mature transcriptomic profile (*Figure 3A*). The latter group includes several members of a vastly expanded family of Fibrinogen-related proteins (FREPs; *Supplementary file 1*), which have been implicated in innate immunity across various phyla (*Zhou et al., 2024*) and may function as venom components. In the 'shared' gene set, most genes are associated with the mature cnidocyte profile and overlap with various neuroglandular subtypes (*Figure 3—figure supplement 1*). This observation supports the hypothesis that the cnidocytes arose from an ancestral neuronal population (*Richards and Rentzsch, 2014*; *Tournière et al., 2020*). We then calculated a gene module score for each gene set to estimate specificity across the dataset, summarizing the specificity of each gene set (*Figure 3B*). Further examination of the gene lists reveals cnidocyte specificity of nematogalectins and minicollagens that serve as structural components for cnidocysts (*Hwang et al., 2010*; *Kurz et al., 1991*). Minicollagens, which have served as important phylum-specific genes (*Holland et al., 2011*), comprise a short collagen domain (about 15 Gly-X-Y repeats) flanked by proline-rich regions and terminal cysteine-rich domains (CRDs; *David et al., 2008*). We identified all five previously described *Nematostella* minicollagen sequences (*Steger et al., 2022*; *Zenkert et al., 2011*) (NvCol-1,–3, −4,–5, –6) along with four additional proteins with incomplete minicollagen sequence features, which we termed 'minicollagen-like' (*Figure 2C*). Intriguingly, we also identified a protein that combines features of both minicollagens and extended ECM-type collagens, suggesting a possible evolutionary origin of minicollagens from this gene family. This protein, NvNCol-7, contains a minicollagen pro-peptide sequence, proline-rich regions, and canonical N- and C-terminal CRDs (*Tursch et al., 2016*). Unlike previously described minicollagens, it includes an extended discontinuous collagen sequence of ~1000 residues, comprising 25 alternating blocks of mostly 12 or 15 Gly-X-Y repeats. These blocks are interrupted by either a single alanine or MPP/SPPSPP sequences, resembling degenerated collagen triplets. The presence of a minicollagen pro-peptide in this collagen suggests its expression in the cnidocyte lineage and secretion into the nematocyst vesicle as a structural component of cnidocyst walls or tubules (*Adamczyk et al., 2010*; *Garg et al., 2023*). This is confirmed by the single cell expression data, which show prominent and exclusive expression in cnidocyte lineages (*Figure 2—figure supplement 2*, *Figure 3—figure supplement 1*). Interestingly, *Nematostella* minicollagens exhibit differential expression across different cnidocyte subtypes, nematocytes, and spirocytes (*Figure 3C*). Spirocytes express NvCol-5 and NvNcol-like-3 and 4, while NvCol-1, 4, and 6 and NvNCol-7 are expressed within the nematocytes and NvCol-3 is expressed in both. Other cnidocyte-specific genes also show differential paralog expression between cnidocyte types, including the four NOWAs (*Engel et al., 2002*; *Garg et al., 2023*), and the vWFA domain proteins (*Figure 3C*). NvTSR-2 vs NvTSR-3 distinguish between the two nematocyte lineages, nem.1 and nem.2. These are postulated to be basitrichous haplonemas/isorhizas based on abundance and distribution across the adult tissue libraries, although these identities have not been validated. In summary, a significant fraction (32%) of matrisomal genes (176 of 551, *Supplementary file 6*) are expressed within the cnidocyte lineage. The proportion of cnidarian-specific factors in this subset is increased (85 of 176, 48%) as compared to the full matrisome (234 of 551, 42%, *Supplementary file 1*), supporting the notion that the cnidocyte proteome represents a specialized ECM adapted to the unique assembly process and biophysical requirements of the cnidarian stinging organelle (*Balasubramanian et al., 2012*; *Ozbek, 2011*).

## Larva-to-polyp transition is marked by factors of basement membrane remodeling and Wnt/PCP signaling

Larva-polyp morphogenesis in *Nematostella* involves significant changes in body shape including the elongation of the body axis and the development of oral tentacles and internal mesenteries (*Stokkermans et al., 2022*). We performed quantitative proteomics using tandem mass tag labeling (TMT; see *Figure 4—figure supplement 1* for normalization steps) to examine whether this process is accompanied by stage-specific variations of matrisome components. As shown above, 38% (210 of 551) of the matrisomal factors were detected by our mass spectrometry analysis (*Figure 1B*). We identified stage-specific mesoglea components by assigning hits (>twofold change and <0.05 false discovery rate [fdr])

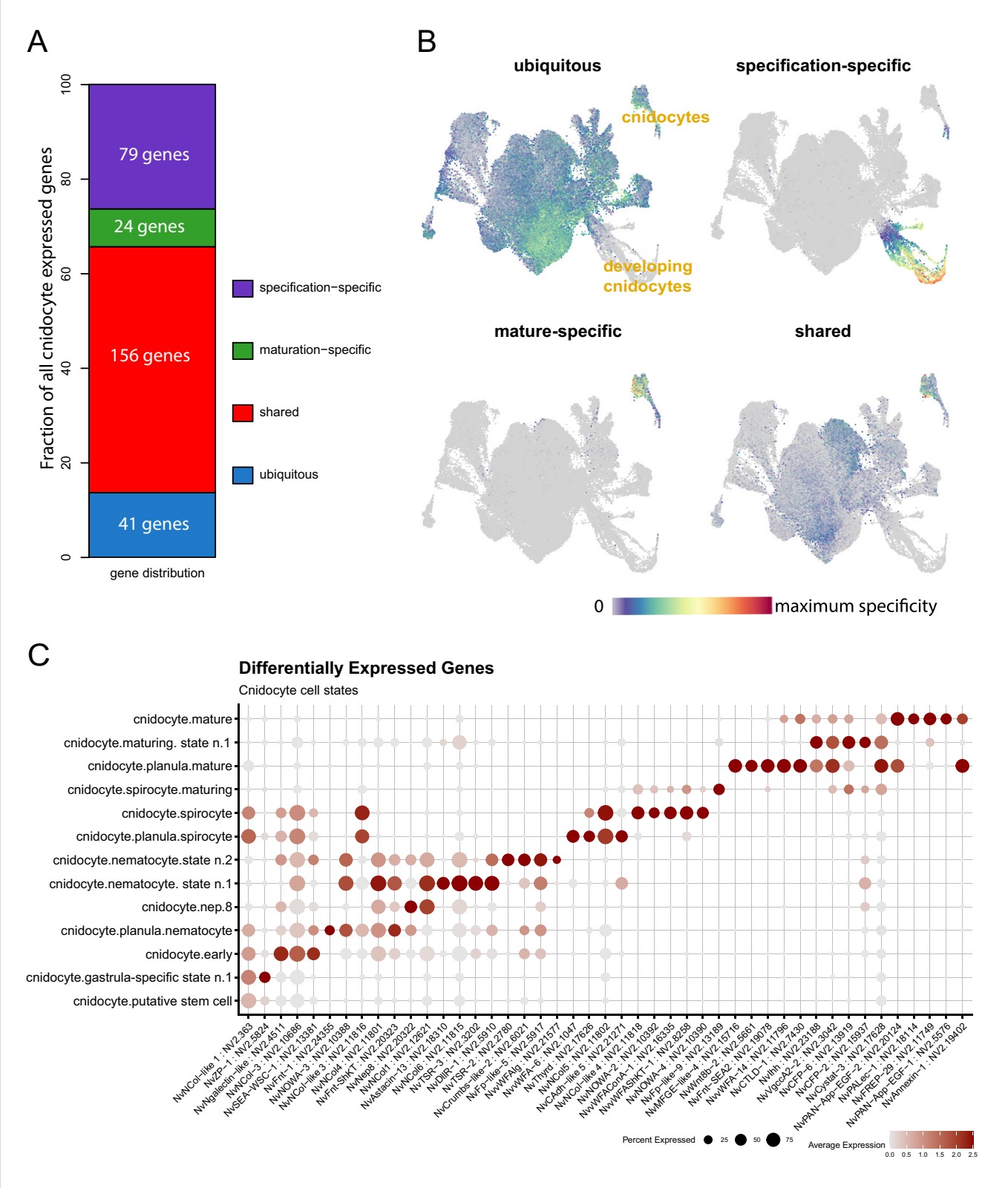

**Figure 3.** Cell-type specificity of cnidocyte-expressed ECM genes. (**A**) The distribution of cnidocyte-expressed genes categorized as 'ubiquitous' (blue: 41), 'shared' (red: 27), 'mature-specific' (green: 38), or 'specification-specific' (purple: 88). (**B**) Expression of the module scores of each gene subset across the main tissue-type data partitions, illustrated on UMAP dimensional reduction. (**C**) Sequential gene expression activation illustrated on a dot

*Figure 3 continued on next page*

Figure 3 continued

plot of top 5 differentially expressed genes (p-value ≤ 0.001) for each cnidocyte cell state. Nematocyte specification shares many genes, while spirocyte specification uses a distinct gene set. Nep.8, nematocyst-expressed protein 8 categorized as venom protein.

The online version of this article includes the following figure supplement(s) for figure 3:

**Figure supplement 1.** Single-cell atlas of cnidocyte-expressed matrisome genes.

using a modified t-test limma for each of the three life stage comparisons (*Supplementary file 7*). Globally, collagens and basement membrane proteoglycans constitute the bulk of the ECM in the core matrisome across all life stages, while secreted factors are the most abundant subcategory among matrisome-associated proteins (*Figure 4A*). As illustrated in the heat map in *Figure 4B*, the transition from larvae to primary polyp is characterized by a general increase of mesoglea components. 94 proteins were differentially upregulated in primary polyps while only four, including vitellogenin and its receptor that are crucial for lipid transport from the ECM into the oocyte (*Lebouvier et al., 2022*), were differentially abundant in larvae. The proteins enriched in primary polyps as compared to larvae include, in addition to various factors involved in general cell adhesion (e.g. cadherin-1, coadhesin-like proteins), two major functional groups (*Figure 4C*, *Supplementary file 7*): (1) factors involved in BM establishment and remodeling and (2) components of the Wnt/PCP signaling pathway. Both groups are indicative of a massive epithelial rearrangement, rather than cell proliferation, as a driver of larva-to-polyp transition (*Stokkermans et al., 2022*). Notably, in addition to laminins, the BM proteoglycan perlecan and numerous astacin, ADAMTS, and MMP family metalloproteases (see *Figure 4—figure supplement 2* for an overview), the first group contained all members of an unusually expanded Polydom protein family (*Figure 4E*). Polydom/SVEP1 is a secreted multi-domain ECM protein initially discovered in a murine bone-marrow stromal cell line (*Gilgès et al., 2000*). In humans, it is composed of eight different domains including an N-terminal vWFA domain, followed by a cysteine-rich ephrin receptor motif, Sushi/CCP and hyalin repeat (HYR) units, EGF-like domains, a central pentraxin domain, and a long tail of 28 Sushi domains terminating in three EGF repeats (*Figure 4E*). Polydom has recently been shown to be a ligand for the orphan receptor Tie1 and induce lymphatic vessel remodeling via PI3K/Akt signaling (*Sato-Nishiuchi et al., 2023*). Earlier studies have shown basement membrane deposition of Polydom and a role in epithelial cell-cell adhesion via integrin binding (*Samuelov et al., 2017*; *Sato-Nishiuchi et al., 2012*). The *Hydractinia* homolog has been characterized as a factor specific to i-cells and to be potentially involved in innate immunity (*Schwarz et al., 2008*). In comparison to vertebrate Polydoms, *Hydractinia* Polydom contains additional Pan/Apple, FA58C, and CUB domains, but has a reduced number of terminal Sushi repeats (*Figure 4E*). In our study, we identified a *Nematostella* Polydom homolog (NvPolydom1) that shares high similarity with the *Hydractinia* and *Hydra* proteins, suggesting a conserved arrangement of domains in cnidarians (*Figure 4E*). This includes a pentraxin-PAN/Apple-FA58C core structure, a tail consisting of six Sushi repeats, and 1–2 terminal CUB domains. Two additional paralogs, NvPolydom2 and NvPolydom3, exhibit differences in their central region, with NvPolydom2 lacking the PAN/Apple domain and NvPolydom3 lacking both PAN/Apple and FA58C domains. Furthermore, we discovered four shorter Polydom-like sequences that share a common EGF-Sushi-HYR-TKE structure but lack vWFA and Pentraxin domains, as well as the terminal Sushi repeats. These Polydom-like proteins, resembling a truncated N-terminal part of canonical Polydoms, possess additional domains at the N- or C-termini, such as thrombospondin type-1 repeat (TSR) or Ig-like domains. The shortest member, referred to as Polydom-related, lacks EGF-like modules and consists of the central Sushi, HYR, and TKE domains found in all Polydom-like sequences, suggesting that this core motif may be essential for biological function. All Polydom and Polydom-like paralogs are expressed within the putative digestive cell state GD.1, with two of these showing additional expression within the uncharacterized secretory cell type S2.tll.2&3 (*Figure 4—figure supplement 3*). This, together with the exceptionally high isoform diversity, indicates a requirement for genetic robustness to account for perturbations of the developmental process regulated by the *Nematostella* Polydoms. Given that mouse Polydom is essential for endothelial cell migration in a Tie-dependent manner (*Sato-Nishiuchi et al., 2023*), it is plausible that the *Nematostella* homologs could serve a similar function for rearrangement of epithelial cells along the BM during primary polyp morphogenesis. Interestingly, the top differentially abundant factor in the primary polyp mesoglea is a secreted integrin-alpha-related protein (sIntREP) containing three integrin-alpha N-terminal domains followed by a stretch of EGF repeats. It is an

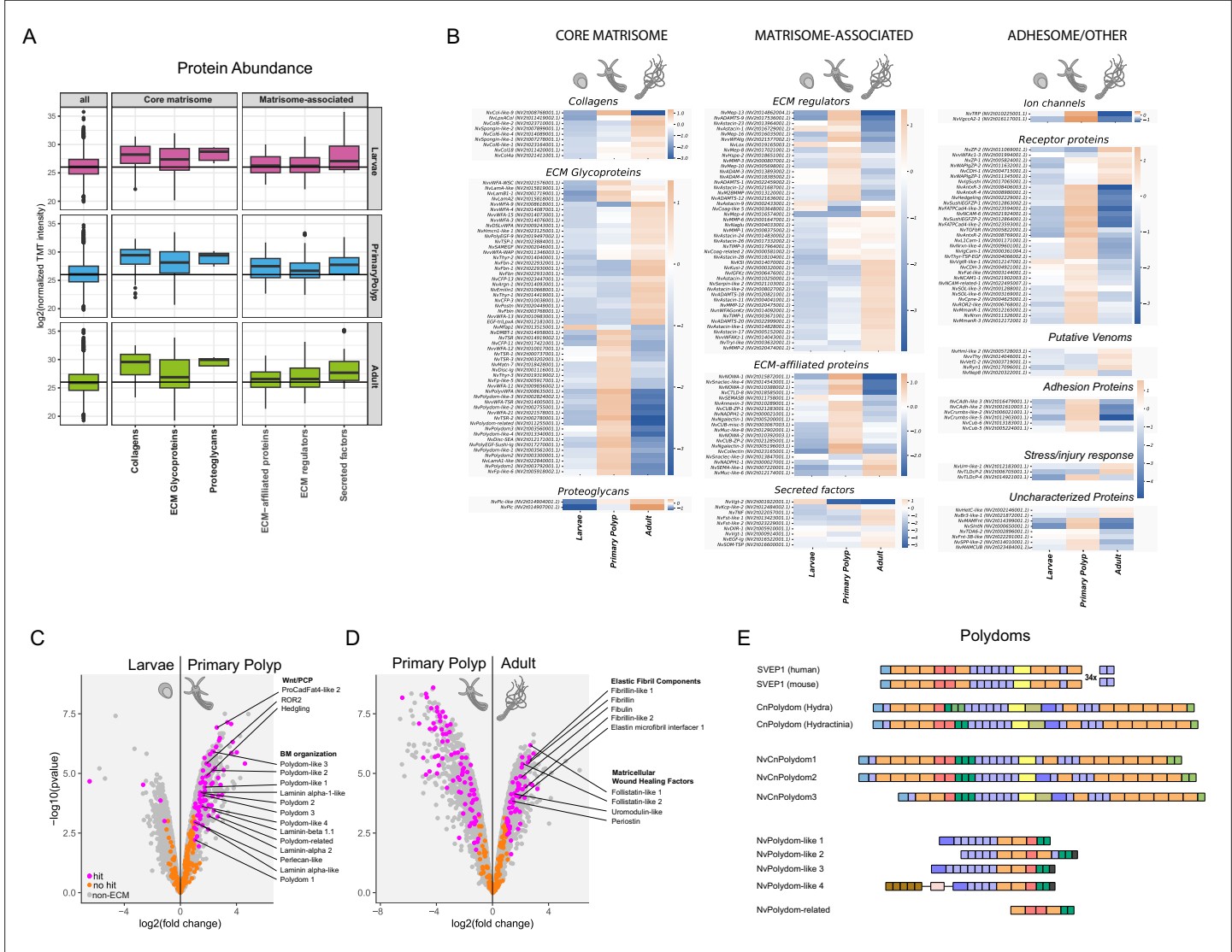

**Figure 4.** Mesoglea dynamics across life stages assessed by quantitative proteomics of isolated mesoglea. (**A**) Boxplot representation of normalized log2 TMT reporter ion intensities for different protein subgroups of the matrisome. 'All' represents all proteins in each respective dataset. A horizontal line indicates median TMT intensity in the complete dataset. (**B**) 2-log transformed median abundances of proteins across different life stages. The curated ECM proteins were filtered for proteins with a twofold change in any of the life stages and a false discovery rate of 0.05 using a moderated t-test (limma). The heatmap shows the 2-log transformed median abundance of 4 samples per life stage. Most proteins are upregulated in only one of the life stages. Notably, BM factors including all polydoms are upregulated in the primary polyp. Most ECM protein categories can be clearly divided into adult-specific and primary polyp-specific proteins underscoring the differential composition of the mesoglea at different life stages. (**C, D**) Volcano plots showing the differential abundance of proteins in the mesoglea extracts of the three different life stages. (**C**) Proteins involved in BM organization including all polydoms and in Wnt/PCP signaling are upregulated during larva-to-primary polyp transition as highlighted. (**D**) The adult mesoglea compared to primary polyps is characterized by an enrichment of elastic fibril components and matricellular glycoproteins involved in wound response and regeneration. gray = non-matrimonial background, orange = insignificant, magenta = differentially abundant matrisome proteins. (**E**) Domain organization of bilaterian and cnidarian polydoms. The *Nematostella* matrisome contains an expanded group of polydoms and polydom-like proteins, including three cnidarian-type polydom paralogs, four shorter polydom-like sequences, and a polydom-related protein, which contains only the core Sushi-HYR-TKE motif. Domain symbols: vWFA (light blue), EGF-like (purple), Sushi/SCR/CCP (orange), Hyalin repeat (red), Pentraxin (yellow), CUB (light green), Tyrosine-protein kinase ephrin (dark green), PAN/Apple (olive green), Ricin B-like (pink), Thrombospondin type-1 repeat (brown), Coagulation factor 5/8 (dark purple), Ig-like (dark grey).

The online version of this article includes the following figure supplement(s) for figure 4:

**Figure supplement 1.** Mesoglea protein abundance overview in specific developmental stages.

**Figure supplement 2.** Domain organization of metalloprotases in the *Nematostella* matrisome.

**Figure supplement 3.** Expression profile of *Nematostella* polydoms.

attractive hypothesis that sIntREP modulates integrin-dependent cell adhesion by Polydoms and other factors to facilitate cell migration. The second group of proteins that contains several components of the Wnt/PCP pathway, including ROR2, protocadherin Fat4-like, and hedgling, further supports the assumption of epithelial cell migration as a main driver of primary polyp morphogenesis. Wnt/PCP signaling, originally described in *Drosophila melanogaster* (*Adler, 2002*; *Gubb and García-Bellido, 1982*), has a well-established role in convergent extension movements of cells during gastrulation to facilitate the elongation of the embryo along its oral-aboral axis (*Gao, 2012*). Its 'core' factors include the transmembrane proteins Frizzled, Van Gogh/Strabismus, Flamingo, and the intracellular components Dishevelled, Prickle, and Diego (*Simons and Mlodzik, 2008*). An additional level of PCP within tissues is regulated by the unusually large protocadherins Fat and Dachsous that interact in a heterophilic manner and display cellular asymmetries (*Matakatsu and Blair, 2004*). Recently, the *Hydra* Fat-like homolog has been reported to be polarized along the oral-aboral axis and to organize epithelial cell alignment via organization of the actin cytoskeleton (*Brooun et al., 2020*). Hedgling is an ancestral, non-bilaterian member of the cadherin superfamily with high similarity to FAT and Flamingo cadherins (*Adamska et al., 2007*). It has a gastrodermal expression in the *Nematostella* primary polyp (*Adamska et al., 2007*), which it shares with Wnt5a and ROR2 (*Supplementary file 4*). ROR2, a highly conserved receptor tyrosine kinase, is the principal transducer of PCP signaling via Wnt5a in the *Xenopus* embryo (*Hikasa et al., 2002*; *Schambony and Wedlich, 2007*). It has also been reported to induce directional cell movements in mammals (*He et al., 2008*) and to induce filopodia formation as a prerequisite for directed cell migration (*Nishita et al., 2006*). Taken together, the molecular dynamics of the mesoglea revealed by quantitative mass spectrometry suggest that massive epithelial rearrangement and directed cell migration underlie axial elongation during primary polyp morphogenesis. Interestingly, IM components such as fibrillar collagens do not appear to contribute largely to this process. In contrast, the adult mesoglea is significantly enriched in elastic fiber components, such as fibrillins and fibulin. This compositional shift likely adds to the visco-elastic properties (*Gosline, 1971a*; *Gosline, 1971b*) of the growing body column (*Figure 4B and D*, *Supplementary file 7*). In addition, the adult mesoglea contains several matricellular factors associated with different aspects of wound healing in vertebrate organisms (*Cárdenas-León et al., 2022*). These include SPARC-related follistatin domain proteins, uromodulin, and periostin. The *Nematostella uromodulin* gene was previously reported to be highly upregulated in the wound ectoderm, likely contributing to the innate immune response (*DuBuc et al., 2014*). The same study showed a circular upregulation of the MMP inhibitor *NvTIMP* around the wound site. Indeed, while metalloproteases are similarly upregulated in adults and primary polyps, we observed a noticeable increase of diverse classes of protease inhibitors in adults (7 vs 2 in primary polyps), including TIMP, Kunitz and Kazal-type protease inhibitors, as well as thyroglobulin repeat proteins (*Novinec et al., 2006*; *Supplementary file 7*). This is indicative of a high degree of protease activity regulation in tissue morphogenesis during growth, as also observed during *Nematostella* whole-body regeneration (*Schaffer et al., 2016*). Taken together, the transition from the primary polyp to the adult is characterized by an increase of elastic fibrillar IM components contributing to the long-range elasticity and resilience of the mesoglea, and a recruitment of wound response factors that together with a complex network of proteases and protease inhibitors likely regulate growth, organogenesis, and tissue differentiation.

## Discussion

The evolution of the ECM, a complex proteinaceous network that connects cells and organizes their spatial arrangement in tissues, has been a key innovation driving the emergence of multicellular life forms (*Brunet and King, 2017*; *Rokas, 2008*). Although ctenophores have recently been identified as the sister group to all other animals (*Schultz et al., 2023*), the hitherto available genomic evidence suggests that they possess only a minimal repertoire of conserved ECM and ECM-affiliated proteins, limiting comparative studies with bilaterians (*Draper et al., 2019*). In contrast, cnidarian genome data have offered broad evidence for conserved matrisomes anticipating the complexity of mammalian species (*Tucker and Adams, 2014*). Here, we identified 551 ECM proteins that comprise the matrisome of the sea anemone *Nematostella vectensis* through in silico prediction using transcriptome databases and analyzed the dynamics of 287 ECM factors by TMT labeling and LC-MS/MS of decellularized mesoglea samples from different life stages. Utilizing cell-type-specific atlases, we showed that the inner gastroderm is the major source of the *Nematostella* ECM, including all 12 collagen-encoding

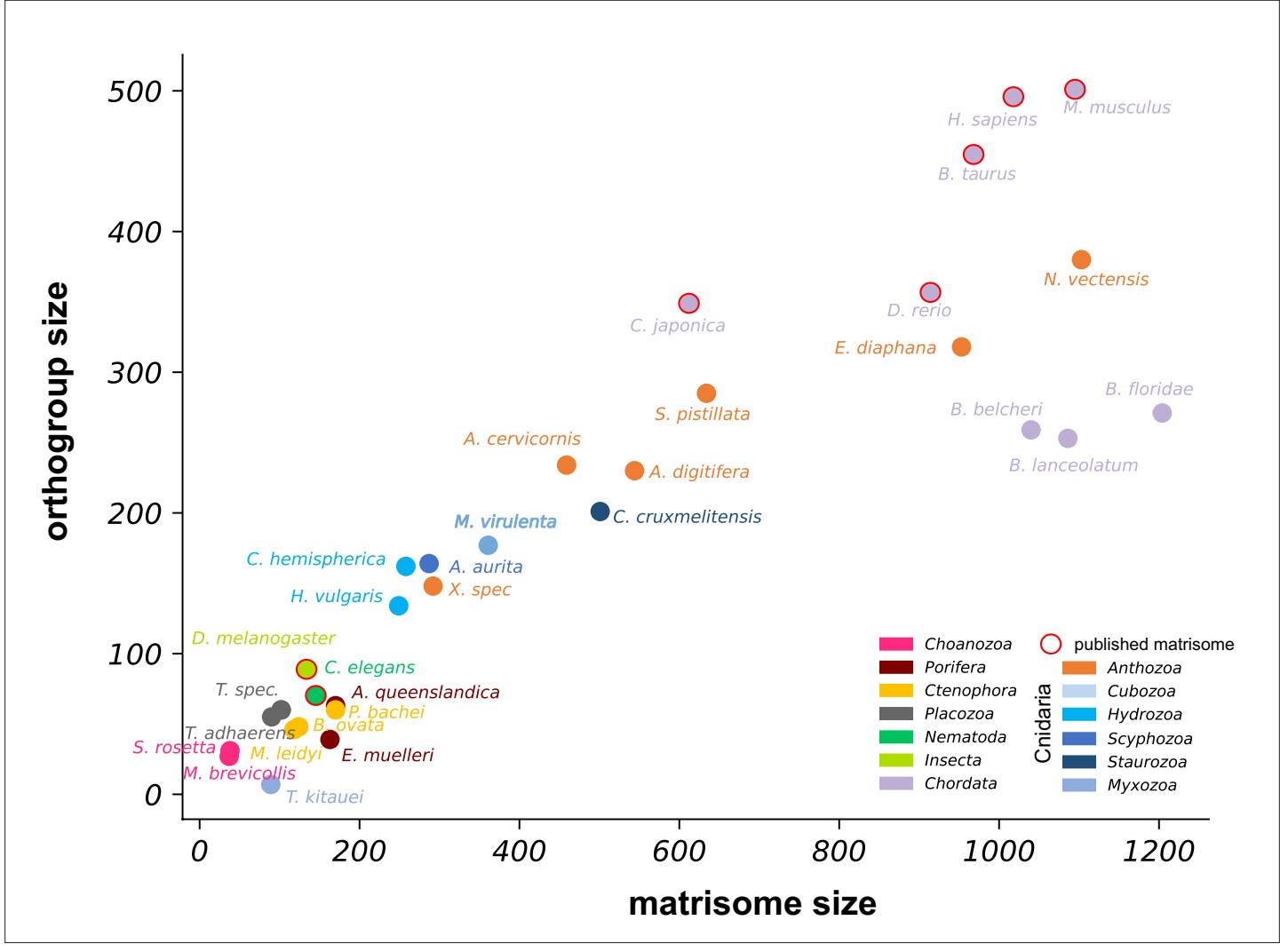

**Figure 5.** Matrisome complexity across metazoan phyla. Matrisome sizes of published and newly generated in silico matrisomes of representative cnidarians and other metazoan species were plotted against their respective orthogroup count. Only proteins from orthogroups shared with at least one published matrisome were counted. Anthozoans generally show a higher matrisome complexity than medusozoan species populating a transitory region between bilaterians and non-bilaterians in the evolutionary trajectory.

genes. This finding supports the model of germ layer evolution proposed by *Steinmetz et al., 2017* where, based on both transcription factor profiles and structural gene sets, the cnidarian inner cell layer (endoderm) is homologous to the bilaterian mesoderm that gives rise to connective tissues. It also contrasts the situation in *Hydra* where both germ layers contribute to the synthesis of core matrisome proteins (*Epp et al., 1986*; *Zhang et al., 2007*). The primacy of the gastrodermis in ECM synthesis might be related to the anthozoan-specific mesenteries, which represent extensions of the mesoglea into the body cavity sandwiched by two endodermal layers. Discrete endo- and ectodermal ECM transcript repertoires would result in a restricted composition of the mesoglea in these mesenteric folds. Whether a gastroderm-based matrisome represents an ancestral state of the cnidarian phylum can only be resolved by the inclusion of omics data from a larger diversity of cnidarian species. While anthozoans such as *Nematostella* have previously been considered a basal group among cnidarians (*Bridge et al., 1992*; *Miller et al., 2000*), more recent phylogenomic studies consider them as a sister clade to medusozoans (*DeBiasse et al., 2024*). To evaluate the complexity of *Nematostella's* matrisome across cnidarians and other metazoan phyla, we plotted matrisome sizes from published databases and newly generated in silico matrisomes of representative species against orthogroup counts (*Figure 5*). For the in silico matrisomes, only orthogroups shared with at

least one found in published matrisomes were counted. Interestingly, anthozoan species generally exhibit a higher complexity than medusozoans and populate a transitory region between bilaterians and non-bilaterians in the evolutionary trajectory. This indicates that the acquisition of complex life cycles in medusozoa, that are distinguished by the pelagic medusa stage, was not accompanied by a diversification of the matrisome repertoire. This is in line with findings from genome data in *Aurelia*, a cnidarian with a medusa stage, that questioned the hypothesis of the medusozoan body plan being derived. Rather, the authors found a redeployment of the existing genetic repertoire (*Gold et al., 2019*). The reduced complexity of the medusozoan ECM might therefore represent a strategy to minimize the cost for ECM remodeling during metamorphosis and rely on a restricted set of conserved genes to form the expanded jellyfish mesoglea.

A significant fraction of *Nematostella's* exceptionally rich matrisome is devoted to the formation of the cnidocyst, a unique cellular novelty of the cnidarian clade (*Babonis et al., 2023*; *Jékely et al., 2015*). Cnidocyst-specific proteins follow an unusual secretion route into the lumen of the growing cnidocyst vesicle, which topologically represents extracellular space (*Ozbek, 2011*). It has therefore been speculated that they originated from neurosecretory vesicles used for predation in early metazoans (*Balasubramanian et al., 2012*). The recent finding that cnidocysts are instrumental for the predatory lifestyle of the *Aiptasia* larvae (*Maegele et al., 2023*) supports the hypothesis of a deeply rooted extrusive mechanism for prey capture in metazoan evolution. In this context, it is intriguing that the majority of *Nematostella* cnidocyte genes shared with other cell types is expressed within neuroglandular subtypes (*Figure 3—figure supplement 1*). In addition, the cnidocyst-specific matrisome contains diverse proteins with repetitive ECM domains that likely have general bioadhesive or fibrous properties that might play a role in entangling and ingesting prey organisms. These include several fibropellin-like and other EGF repeat proteins (poly-EGF) as well as thrombospondin type-1 repeat (TSR) proteins, such as properdin-likes.

The changes in the matrisome profiles across *Nematostella's* major life stages suggest a highly dynamic epithelial rearrangement during primary polyp morphogenesis. The involvement of Wnt/PCP factors in this process indicates similar cell migration and reorientation events as during convergent extension processes in gastrulation. Kumburegama et al. have shown that primary archenteron invagination and apical constriction of bottle cells in *Nematostella* is dependent on the PCP components *strabismus* (*Kumburegama et al., 2011*) and *Fzd10* (*Wijesena et al., 2022*). Mesentery formation in primary polyps, which involves sequential folding events of the endodermal epithelium (*Berking, 2007*), likely involves similar molecular pathways. As already observed by *Appelöf, 1900*, mesoglea synthesis follows invagination during this process. The molecular network composed of Wnt/PCP and basal membrane factors that our data revealed might therefore indicate an actomyosin-controlled invagination of the endodermal layer followed by BM production to re-align the cells in their apico-basal architecture. The upregulation of wound response factors in the adult animal might indicate a transient loss of tissue integrity during collective cell migration, which could entail an actin-based purse-string mechanism as during wound healing (*Begnaud et al., 2016*; *Bischoff et al., 2021*). Future work will further decipher the gene-regulatory network controlling polyp morphogenesis in this anthozoan model.

# Materials and methods

## Key resources table

| Reagent type (species) or resource | Designation | Source or reference | Identifiers | Additional information |
|---|---|---|---|---|
| Biological sample (*Nematostella vectensis*) | Larvae (3 dpf), primary polyps (10 dpf), adults (≥1 year) | Mark Q. Martindale, Whitney Lab; *Putnam et al., 2007* | | Cultured in lab conditions; used for mesoglea isolation, immunostaining, proteomics. Mixed sex animals used in all experiments |
| Antibody | Anti-Laminin (rabbit polyclonal) | This paper | | Custom polyclonal antibody produced by eurogentec, epitope in Nv Laminin γ1 chain (1:100 for IF, 1:2 for EM) |
| Antibody | Anti-Collagen IV (guinea pig polyclonal) | This paper | | Custom polyclonal antibody produced by eurogentec, epitope in NvCol4b (1:10 for EM) |

*Continued on next page*

*Continued*

| Reagent type (species) or resource | Designation | Source or reference | Identifiers | Additional information |
|---|---|---|---|---|
| Antibody | Anti-Collagen II-like (rat polyclonal) | This paper | | Custom polyclonal antibody produced by eurogentec, epitope in NvCol2c (1:10 for EM) |
| Antibody | Anti-Pan-Collagen (rat polyclonal) | This paper | | Custom polyclonal antibody produced by eurogentec, consensus fibrillar collagen motif (1:100 for IF, 1:2 for EM) |
| Antibody | Alexa Fluor 488 goat anti-rat IgG (H+L) (goat polyclonal) | Thermo Fisher Scientific | Cat# A-11006 RRID:AB_2534074 | Secondary antibody (1:400) |
| Antibody | Alexa Fluor 568 goat anti-rabbit IgG (H+L) (goat polyclonal) | Thermo Fisher Scientific | Cat# A-11011 RRID:AB_143157 | Secondary antibody (1:400) |
| Antibody | Goat anti-rabbit IgG (10 nm colloidal gold) (goat polyclonal) | British Biocell | Cat# EM.GAR10/1 RRID:AB_2715527 | Immunogold EM (1:150) |
| Antibody | Goat anti-rat IgG (10 nm colloidal gold) (goat polyclonal) | British Biocell | Cat# EM.GAT10/1 RRID:AB_2715527 | Immunogold EM (1:150) |
| Antibody | Nanogold-IgG Goat anti-Guinea Pig IgG (goat polyclonal) | Nanoprobes | Cat# 2054 RRID:AB_3711173 | Immunogold EM (1:150) |
| Antibody | Nanogold-IgG Goat anti-Rat IgG (H+L) (goat polyclonal) | Nanoprobes | Cat# 2007 RRID:AB_3711173 | Immunogold EM (1:150) |
| Chemical compound, drug | Chymotrypsin, sequencing grade | Promega | Cat# V1061 | Mass spectrometry |
| Chemical compound, drug | Cysteine | Sigma-Aldrich | Cat# C7352 | Egg dejellying |
| Chemical compound, drug | DAPI | Sigma-Aldrich | Cat# D9542 | Nuclear stain |
| Chemical compound, drug | Dithiothreitol | Sigma-Aldrich | Cat# D9779 | Mesoglea preparation |
| Chemical compound, drug | EGTA | Sigma Aldrich | Cat# 324626 | Electron microscopy |
| Chemical compound, drug | Formaldehyde solution min. 37% | Merck KGaA | Cat# 252549 | Electron microscopy |
| Chemical compound, drug | Glutaraldehyde | Sigma-Aldrich | Cat# G5882 | Electron microscopy |
| Chemical compound, drug | HEPES | Sigma-Aldrich | Cat# H3375 | Mesoglea preparation |
| Chemical compound, drug | HQ-Silver | Nanoprobes Yaphank | Cat# 2012 | Electron microscopy |
| Chemical compound, drug | Magnesium chloride | Sigma-Aldrich | Cat# M8266 | Immunocytochemistry |
| Chemical compound, drug | N-lauryl-sarcosinate | Sigma-Aldrich | Cat# L5125 | Mesoglea decellularization |
| Chemical compound, drug | OASIS HLB µElution Plate | Waters | Cat# 186001828BA | Mass spectrometry |
| Chemical compound, drug | Osmium tetroxide | Sigma Aldrich | Cat# O5500 | Electron microscopy |
| Chemical compound, drug | Precast 4–12% gradient gels | Carl Roth | Cat# 3673.2 | SDS-PAGE |
| Chemical compound, drug | TMT10plex Isobaric Label Reagent | ThermoFisher | Cat# 90110 | Mass spectrometry |
| Chemical compound, drug | Trichloroacetic acid | Sigma-Aldrich | Cat# T6399 | Mesoglea preparation |
| Chemical compound, drug | Triton X-100 | Sigma-Aldrich | Cat# T8787 | Immunocytochemistry |

*Continued on next page*

*Continued*

| Reagent type (species) or resource | Designation | Source or reference | Identifiers | Additional information |
|---|---|---|---|---|
| Chemical compound, drug | Tween-20 | Roche | Cat# 11332465001 | Immunocytochemistry |
| Chemical compound, drug | Uranyl acetate | Electron Microscopy Sciences | Cat# 541-09-3 | Electron microscopy |
| Software, algorithm | BLAST | *Altschul et al., 1990* | RRID:SCR_004870 | Database searches |
| Software, algorithm | Custom Python scripts | This paper | RRID:SCR_024202 | For domain and orthogroup analysis |
| Software, algorithm | DeepLoc-2 | *Thumuluri et al., 2022* | RRID:SCR_026503 | Protein localization prediction |
| Software, algorithm | Fiji (ImageJ) | *Schindelin et al., 2012* | RRID:SCR_003070 | Image processing |
| Software, algorithm | IsobarQuant | *Franken et al., 2015* | RRID:SCR_016732 | MS data analysis |
| Software, algorithm | limma (R package) | *Ritchie et al., 2015* | RRID:SCR_010943 | Proteomics analysis |
| Software, algorithm | Mascot | Matrix Science | RRID:SCR_014322 | MS data analysis |
| Software, algorithm | NIS elements Imagine software | Nikon Instruments Inc. | | Image processing. https://www.microscope.healthcare.nikon.com/products/software/nis-elements |
| Software, algorithm | OrthoFinder v2.5.4 | *Emms and Kelly, 2019* | RRID:SCR_017118 | Orthogroup prediction |
| Software, algorithm | Seurat package | *Stuart et al., 2019* | RRID:SCR_016341 | Single cell expression analysis |
| Software, algorithm | SignalP-6.0 | *Teufel et al., 2022* | RRID:SCR_015644 | Signal peptide prediction |
| Software, algorithm | SMART | *Schultz et al., 1998* | RRID:SCR_005026 | Protein domain analysis |
| Other | Single-cell RNA atlas | *Cole et al., 2024* | | Expression data |
| Other | Nikon A1R Confocal Laser Scanning Microscope | Nikon, Tokyo, Japan | RRID:SCR_020317 | Confocal microscopy |
| Other | Nikon Eclipse 80i microscope | Nikon, Tokyo, Japan | RRID:SCR_015572 | Fluorescence microscopy |
| Other | Gemini C18 column (3 μm, 110 Å, 100 × 1.0 mm) | Phenomenex | Cat# 00D-4439-A0 | Mass spectrometry |
| Other | Agilent 1200 Infinity high-performance liquid chromatography system | Agilent | RRID:SCR_018018 | Mass spectrometry |

## Nematostella culture

Animals used in this study were sea anemones of mixed sex and originally obtained from Mark Q. Martindale, The Whitney Laboratory for Marine Bioscience (*Putnam et al., 2007*). For all experiments, animals were randomly selected. Adult *Nematostella* were kept in plastic boxes at 18°C in 1/3 artificial sea water (~11 ppt; *Nematostella* medium) in the dark. They were fed with freshly hatched *Artemia* nauplii and cleaned once per week. To induce spawning, animals were transferred to 27°C *Nematostella* medium in light for 8 hr and then washed with 18°C *Nematostella* medium. The egg patches were collected and dejellied in 5% cysteine solution for 15 min. Unless otherwise stated, the embryos were left to develop in *Nematostella* medium at room temperature (RT) in normal day/night cycles.

## Immunocytochemistry

Larvae on 3 dpf, primary polyps on 10 dpf, and small adult polyps were collected and left to relax at 27°C in direct light for 30 min. A solution of 7% $MgCl_2$ in seawater was slowly added, and the animals were anesthetized for 20 min. Fixation was performed with Lavdovsky's fixative (50% ethanol, 36% $H_2O$, 10% formaldehyde, 4% acetic acid) for 30 min at RT after which the samples were incubated in 150 mM Tris, pH 9.0, 0.05% Tween-20, for 10 min. An incubation step at 70°C for 10 min and a subsequent cooling to RT was followed by three 10 min washing steps in PBS, PBS, 0.1% Tween-20, and PBS, 0.1% Triton-X100, respectively. Primary antibody incubation (rabbit anti-Laminin, rat anti-Pan-Collagen) was performed at 1:100 in 0.5% milk powder overnight at 4°C. The samples were washed 3 times for 10 min in PBS, 0.1% Tween-20, and incubated with secondary antibodies (Alexa Fluor 488 goat anti-rat IgG (H+L), Thermo Fisher Scientific; Alexa Fluor 568 goat anti-rabbit IgG (H+L), Thermo

Fisher Scientific) at 1:400 for 2.5 hr at RT. Prior to mounting on object slides with Mowiol, DAPI was added at 1:1000 for 30 min. Decellularized mesogleas were transferred onto a microscopy slide lined with liquid blocker. The mesogleas of larvae and primary polyps were carefully stuck to the slide using an eyelash. The staining protocol followed the whole mount immunocytochemistry protocol with 10 min fixation and only one washing step per wash to avoid washing off of the mesogleas. The antibodies were incubated on the slides in a Petri dish with a wet paper towel to prevent evaporation. Images were acquired with an A1R microscope at the Nikon Imaging Facility Heidelberg. Further image processing was performed with Fiji ImageJ v1.53t.

## Mesoglea decellularization

All samples were prepared as biological triplicates. About 500,000 larvae (3 dpf) and primary polyps (10 dpf), and four adult animals were collected. The adult polyps were cut open along the oral-aboral axis using a scalpel to ease the decellularization of the endoderm. All samples were incubated in 0.5% N-lauryl-sarcosinate for 5 min and then frozen in liquid nitrogen. After thawing at RT, the samples were transferred to ddH$_2$O using a 70 µm sieve for larvae and primary polyps. The mesogleas were decellularized in ddH$_2$O by repeated pipetting with a flamed glass pipette and frequent water changes. The progress of decellularization was checked repeatedly by phase contrast microscopy at 60x magnification using a Nikon 80i microscope. As a final quality control, a few sample mesogleas were stained with DAPI for 10 min in PBS to visualize residual cells or nematocysts. Decellularized mesogleas were then picked individually for further analysis. For SDS-PAGE analysis, isolated mesogleas (40 µg) were dissolved in 1 M dithiothreitol (DTT), boiled at 90°C for 2 hr, and loaded on a precast 4–15% gradient gel (Carl Roth) after a quick spin.

## Sample preparation for SP3 and TMT labeling

Isolated mesogleas (larvae and primary polyps, N=150, adults, N=4) were dissolved in 1 M DTT for 30 min at 90°C and protein extraction was performed using trichloroacetic acid (TCA) precipitation. TCA pellets were resuspended in 50 µL 1% SDS, 50 mM HEPES pH 8.5. Reduction of disulfide bridges in cysteine-containing proteins was performed with DTT (56°C, 30 min, 10 mM in 50 mM HEPES, pH 8.5). Reduced cysteines were alkylated with 2-chloroacetamide (RT, in the dark, 30 min, 20 mM in 50 mM HEPES, pH 8.5). Samples were prepared using the SP3 protocol (*Hughes et al., 2014*; *Hughes et al., 2019*) and trypsin was added in an enzyme to protein ratio of 1:50 for overnight digestion at 37°C. Then, peptide recovery was performed in HEPES buffer by collecting the supernatant on a magnet and combining it with the second elution wash of beads with HEPES buffer. Peptides were labeled with TMT10plex (*Werner et al., 2014*) Isobaric Label Reagent according to the manufacturer's instructions. For further sample clean up, an OASIS HLB µElution Plate was used for each sample separately. A control run was performed to be able to mix equal peptide amounts based on the MS signal in each run, and samples were combined for TMT9plex accordingly. Offline high pH reverse phase fractionation was carried out on an Agilent 1200 Infinity high-performance liquid chromatography system, equipped with a Gemini C18 column (3 µm, 110 Å, 100 × 1.0 mm, Phenomenex; *Reichel et al., 2016*).

## Mass spectrometry and data analysis

Each biological sample was subjected to analysis in technical triplicates. LC-MS/MS, Liquid Chromatography (LC) was performed as previously described for *Hydra* mesoglea (*Veschgini et al., 2023*). IsobarQuant (*Franken et al., 2015*) and Mascot (v2.2.07) were used to process the acquired data, which was searched against the *Nematostella vectensis* NV2 (wein_nvec200_tcsv2) protein models (https://simrbase.stowers.org/starletseaanemone) containing common contaminants and reversed sequences. The following modifications were included in the search parameters: Carbamidomethyl (C) and TMT10 (K) (fixed modification), Acetyl (Protein N-term), Oxidation (M) and TMT10 (N-term) (variable modifications). For the full scan (MS1), a mass error tolerance of 10 ppm and for MS/MS (MS2) spectra of 0.02 Da was set. Further parameters were: trypsin as protease with an allowance of a maximum of two missed cleavages, a minimum peptide length of seven amino acids, and at least two unique peptides were required for a protein identification. An inclusion of lysine and proline hydroxylation as variable modifications did not increase the detection of *bona fide* collagens, prompting us to omit these parameters. The false discovery rate on peptide and protein level was set to 0.01. The raw

output files of IsobarQuant (protein.txt – files) were processed using R. Contaminants were filtered out and only proteins that were quantified with at least two unique peptides were considered for the analysis. 5056 proteins passed the quality control filters. Log2-transformed raw TMT reporter ion intensities ('signal_sum' columns) were first cleaned for batch effects using limma (*Ritchie et al., 2015*) and further normalized using variance stabilization normalization (*Huber et al., 2002*; see *Figure 4— figure supplement 1* for an overview of these steps). Proteins were tested for differential expression using the limma package. The replicate information was added as a factor in the design matrix given as an argument to the 'lmFit' function of limma. A protein was annotated as a hit with a false discovery rate (fdr) smaller than 5% and a fold-change of at least 2. For the heatmap shown in *Figure 4B*, the log2-transformed median abundance of all samples for each life stage was calculated.

## In silico matrisome prediction

Domains for the protein models of the SIMRbase *Nematostella vectensis* NV2 (wein_nvec200_tcsv2) transcriptome (https://simrbase.stowers.org/starletseaanemone) were de novo annotated using InterProScan (*Jones et al., 2014*). The proteins were then filtered positively using a list of known ECM protein domains and negatively by the presence of the respective exclusive domains according to *Naba et al., 2012*. Signal peptides were predicted using SignalP-6.0 (*Teufel et al., 2022*) and DeepLoc-2 (*Thumuluri et al., 2022*). The latter was also used to predict the cellular localization of proteins. In addition, OrthoFinder vers. 2.5.4 was used with default settings to predict orthogroups from all predicted matrisomes and published matrisomes from *M. musculus*, *H. sapiens* (*Naba et al., 2012*), *B. taurus* (*Listrat et al., 2023*), *C. japonica* (*Huss et al., 2019*), *D. melanogaster* (*Davis et al., 2019*), *D. rerio* (*Nauroy et al., 2018*), *S. mediterranea* (*Cote et al., 2019*) and *C. elegans* (*Teuscher et al., 2019*). Finally, all *Nematostella* sequences were manually annotated by comparing their domain architecture to published protein groups. Domain and orthogroup analysis were performed using custom python scripts (See data availability statement). For the orthogroup analysis, the phylogenetically hierarchical orthogroups predicted by OrthoFinder were analyzed. To prevent domain redundancy, we restricted the analysis to SMART domains for the domain comparison, as domain comparisons for other domain databases showed similar results. To achieve a better orthogroup definition, we predicted additional in silico matrisomes for a number of available protein model datasets in non-bilaterian species using the same bioinformatic pipeline as for *Nematostella*: Choanoflagellata: *Monosiga brevicollis*, *Salpingoeca rosetta*; Porifera: *Amphimedon queenslandica*, *Ephydatia muelleri*; Ctenophora: *Mnemiopsis leidyi*, *Pleurobrachia bachei*, *Beroe ovata* (http://ryanlab.whitney.ufl.edu/bovadb), *Pleurobrachia bachei* (*Moroz et al., 2014*); Placozoa: *Tricoplax adhaerens*, *Tricoplax spec*; Cnidaria: *Aurelia aurita* (*Gold et al., 2019*), *Clytia hemispherica* (http://marimba.obs-vlfr.fr), *Exaiptasia diaphana* (*Oakley et al., 2016*), *Hydra vulgaris*, *Acropora digitifera* (*Shinzato et al., 2021*), *Stylophora pistillata*, *Calvadosia cruxmelitensis* (*Ohdera et al., 2019*), *Morbakka virulenta* (*Khalturin et al., 2019*), *Thelohanellus kitauei*, *Acropora digitifera*, *Porites asteroides* (*Kenkel et al., 2013*), *Xenia spec.* (*Hu et al., 2020*), Chordata: *Branchiostoma belcheri*, *Branchiostoma floridae*, *Branchiostoma lanceolatum* (Uniprot Reference Proteomes). To identify orthogroups specific to cnidarians, we filtered the OrthoFinder-derived orthogroups and phylogenetic hierarchies, selecting only those that exhibited cnidarian exclusivity. To identify potential venoms in our matrisome dataset, we performed BLAST searches against the ToxProt database of known animal toxins (*Jungo et al., 2012*). We categorized as 'putative venoms' (*Supplementary file 1*) candidates that exhibited significant matches in this database (E>1e–03).

## Single-cell RNA expression

The expression matrix corresponding to the predicted ECM genes was extracted from the updated single cell atlas (*Cole et al., 2024*). To generate expression data, the dataset was first separated into three life-cycle stages: samples from 18 hr gastrula until 4 d planula were classified as 'larva', samples from 5 d through 16 d primary polyps were classified as 'primary polyp', and all samples derived from juvenile or adult tissues were classified as 'adult'. The full gene matrix was filtered down to only the 829 models corresponding to the curated list of ECM genes. For plotting expression values, the principal tissue-type annotations were further collapsed to cluster together early and late ectodermal clusters, specification and mature cnidocyte states, and to collapse the primary germ cells and putative stem cells into a single data partition. Differentially expressed genes were calculated across all annotated

cell-type states using the Seurat vs.4 function Seurat::FindAllMarkers, requiring a return-threshold of 0.001, and a minimum detection in 20% of any cluster. Module expression scores for different gene sets (core, associated, and other for the full dataset, and 'ubiquitous', 'shared', 'specification-specific' and 'mature-specific' for the cnidocyte subset) were calculated using the Seurat function Seurat::AddModuleScore. The cnidocyte genes were binned as described above according to summarized expression data generated by the Seurat::DotPlot function, in the 'data' matrix of the resulting ggplot.

## Electron microscopy

For morphology, *Nematostella* larvae and primary polyps were processed as previously described for *Hydra* (*Böttger et al., 2012*; *Garg et al., 2023*). Briefly, animals were subjected to cryofixation (high-pressure freezing, freeze-substitution, and epoxy resin embedding: HPF/FS: *Figure 1—figure supplement 3A, B, G, H*) or to standard chemical fixation (glutaraldehyde, followed by OsO4, resin embedding: CF: *Figure 1—figure supplement 3F, L*). Ultrathin sections were optionally stained with uranyl acetate and lead for general contrast enhancement or with periodic acid, thiocarbohydrazide, and silver proteinate to highlight periodic acid-Schiff-positive constituents (*Figure 1—figure supplement 3H*). For Tokuyasu-immunoelectron microscopy (*Tokuyasu, 1973*) samples were either fixed for >3 days at RT with 4 % w/v formaldehyde solution in PHEM (5 mM HEPES, 60 mM PIPES, 10 mM ethylene glycol tetraacetic acid (EGTA), 2 mM $MgCl_2$), pH 7.0 (TOK: *Figure 1—figure supplement 3M, O*) or by using a new modification of established HPF/FS-sample rehydration methods (*Ripper et al., 2008*; *Schmiedinger et al., 2013*); this modification included freeze substitution with methanol containing 3.2 % w/v formaldehyde, 0.08 % w/v uranyl acetate, and 8.8% $H_2O$, removal of uranyl acetate at 4°C (on ice) and partial sample rehydration and postfixation through incubation in Lavdovsky's fixative for 1 hr at RT (HPF/FS/RH-Lav: *Figure 1—figure supplement 3N, P*). Fixed samples were rinsed with PHEM buffer and further processed for thawed cryosection immunogold labeling (*Tokuyasu, 1973*) as previously described (*Garg et al., 2023*). Anti-Lam and anti-PanCol label on standard TOK-sections were visualized with goat anti-rabbit or goat anti-rat secondary antibodies coupled to 10 nm colloidal gold. Anti-Col4 and anti-Col2c labeling was performed on HPF/FS/RH-Lav samples by using Nanogold-IgG Goat anti-Guinea Pig IgG or Nanogold-IgG Goat anti-Rat IgG (H+L), respectively, followed by silver enhancement with HQ silver.

## Acknowledgements

This work was supported by the German Science Foundation (DFG) (Collaborative Research Center 1324 (B07) and OE 416/8–1) to S.Ö. B.G.B. acknowledges the Schmeil Foundation Heidelberg for its support. We thank Prakash Balasubramanian for preparing SDS-PAGE images of the *Hydra* and *Nematostella* mesogleas and Josephine C Adams for fruitful discussions.

## Additional information

### Funding

| Funder | Grant reference number | Author |
| --- | --- | --- |
| Deutsche Forschungsgemeinschaft | Collaborative Research Center 1324 (B07) | Suat Özbek |
| Deutsche Forschungsgemeinschaft | OE 416/8-1 | Suat Özbek |
| Schmeil Foundation Heidelberg | | Bruno Gideon Bergheim |

The funders had no role in study design, data collection and interpretation, or the decision to submit the work for publication.

### Author contributions

Bruno Gideon Bergheim, Software, Formal analysis, Validation, Investigation, Visualization, Methodology, Writing – original draft, Writing – review and editing; Alison G Cole, Frank Stein, Data

curation, Formal analysis, Validation, Investigation, Visualization, Methodology, Writing – original draft, Writing – review and editing; Mandy Rettel, Data curation, Validation, Investigation, Methodology; Stefan Redl, Formal analysis, Investigation, Visualization; Michael W Hess, Resources, Investigation, Visualization, Methodology, Writing – original draft, Writing – review and editing; Aissam Ikmi, Resources, Writing – review and editing; Suat Özbek, Conceptualization, Data curation, Supervision, Funding acquisition, Validation, Writing – original draft, Project administration, Writing – review and editing

#### Author ORCIDs
Alison G Cole ⓘ https://orcid.org/0000-0002-7515-7489
Mandy Rettel ⓘ https://orcid.org/0000-0002-8304-3385
Michael W Hess ⓘ https://orcid.org/0000-0002-5154-3553
Suat Özbek ⓘ https://orcid.org/0000-0003-2569-3942

Reviewer #1 (Public review): https://doi.org/10.7554/eLife.105319.3.sa1
Author response https://doi.org/10.7554/eLife.105319.3.sa2

---

## Additional files

#### Supplementary files
Supplementary file 1. *In silico* matrisome prediction of the *Nematostella* mesoglea including results of differential abundance analysis using moderated t-test (limma) for each developmental comparison (Adult - Larvae, Larvae - Primary Polyp, Adult - Primary Polyp). Related to *Figures 1 and 4*.

Supplementary file 2. Full mass spectrometry data of the *Nematostella* mesoglea. Related to *Figures 1 and 4*. qupm: quantified unique peptide matches, top3: average log10 MS1 intensity of the three most abundant peptides for a given protein. norm_batchcl_raw_signal_sum: normalized and batch-cleaned reporter ion intensities from the TMT experiment.

Supplementary file 3. Raw data of mesoglea and BM thickness as well as fibril thickness measurements in electron micrographs of larva and primary polyp cryosections. Related to *Figure 1—figure supplement 3*.

Supplementary file 4. Gene expression profiles indicating the average expression (avg.exp), the percentage of the cells with gene expression (pct.exp), and the relative expression levels (avg.exp. scaled) for all ECM genes across all cell state identities (cell.state.id) in *Cole et al., 2024*. pSC, putative germ cell, PGC, primary germ cell, NPC, neuroglandular progenitor cell. GD, digestive gland cell, NGD, digestive neuroglandular cell. Related to *Figure 2*.

Supplementary file 5. Single cell expression profiles of core matrisomal genes. Related to *Figure 2*. Output of the FindAllMarkers function of the Seurat package (vs.4.4.0) run with default parameters using all core matrisome genes across all clusters annotated in *Cole et al., 2024* ( cell.state.id). gene: core matrisome gene tested, p_val: calculated p-value of the expression difference; avg_log2FC: calculated average log2 fold change between cluster 1 (cell.state.id) and the rest of the dataset; pct.1: percentage of cells in cluster 1 (cell.state.id) with gene expression; pct.2: percentage of cells within the remaining clusters with gene expression; p_val_adj: adjusted p-value.

Supplementary file 6. Matrix of average expression values for all cnidocyte-specific genes (rows) across all cell state identities in *Cole et al., 2024* (columns). Related to *Figure 3*.

Supplementary file 7. Quantitative mass spectrometry data of mesoglea samples from different life stages. Related to *Figure 4*.

Supplementary file 8. README for *Supplementary files 1–7*.

MDAR checklist

#### Data availability
The MS run results can be found in the PRIDE Database (PXD045345). An R script for generating all single cell RNA expression figures and the sequences and OrthoFinder results of the in silico matrisomes are available at GitHub (copy archived at *Özbek, 2025*). The original single cell atlas is available through the UCSC Cell Browser at https://sea-anemone-atlas.cells.ucsc.edu Identifier: Nv2 Atlas.

The following dataset was generated:

| Author(s) | Year | Dataset title | Dataset URL | Database and Identifier |
|---|---|---|---|---|
| Stein F, Oezbek S | 2025 | Nematostella extracellular matrix proteome of planula, primary polyp and adult | https://www.ebi.ac.uk/pride/archive/projects/PXD045345 | PRIDE, PXD045345 |

The following previously published datasets were used:

| Author(s) | Year | Dataset title | Dataset URL | Database and Identifier |
|---|---|---|---|---|
| Cole AG, Technau U, Steger J | 2022 | A time course of sea anemone development | https://www.ncbi.nlm.nih.gov/geo/query/acc.cgi?acc=GSE200198 | NCBI Gene Expression Omnibus, GSE200198 |
| Cole AG, Technau U, Steger J | 2022 | Sea anemone single cell tissue profiling [scRNA-seq] | https://www.ncbi.nlm.nih.gov/geo/query/acc.cgi?acc=GSE154105 | NCBI Gene Expression Omnibus, GSE154105 |

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
