## [Editor Report · eLife Assessment]

This **valuable** study provides a comprehensive description of the *Nematostella vectensis* matrisome - the genes encoding the proteins of the extracellular matrix. The authors combine new mass spectrometry data with bioinformatic analyses of previously published genomic and single-cell RNAseq data. The analysis is thorough, and the discussion and conclusions are **convincing**. This work will be of interest to biologists working on the evolution of the matrisome, as well as more broadly those working with non-bilaterian animals.

---

## [Referee Report · Reviewer #1 (Public review)]

Summary:

In this manuscript entitled "Molecular dynamics of the matrisome across sea anemone life history", Bergheim and colleagues report the prediction, using an established sequence analysis pipeline, of the "matrisome" - that is, the compendium of genes encoding constituents of the extracellular matrix - of the starlet sea anemone *Nematostella vectensis*. Re-analysis of an existing scRNA-Seq dataset allowed the authors to identify the cell types expressing matrisome components and different developmental stages. Last, the authors apply time-resolved proteomics to provide experimental evidence of the presence of the extracellular matrix proteins at three different stages of the life cycle of the sea anemone (larva, primary polyp, adult) and show that different subsets of matrisome components are present in the ECM at different life stages with, for example, basement membrane components accompanying the transition from larva to primary polyp and elastic fiber components and matricellular proteins accompanying the transition from primary polyp to the adult stage.

Strengths:

The ECM is a structure that has evolved to support the emergence of multicellularity and different transitions that have accompanied the complexification of multicellular organisms. Understanding the molecular makeup of structures that are conserved throughout evolution is thus of paramount importance.

The in-silico predicted matrisome of the sea anemone has the potential to become an essential resource for the scientific community to support big data annotation efforts and better understand the evolution of the matrisome and of ECM proteins, an important endeavor to better understand structure/function relationships. Toward this goal, the authors provide a comprehensive list with extensive annotations and cross-referencing of the 551 genes encoding matrisome proteins in the sea anemone genome.

This study is also an excellent example of how integrating datasets generated using different -omic modalities can shed light on various aspects of ECM metabolism, from identifying the cell types of origins of matrisome components using scRNA-Seq to studying ECM dynamics using proteomics.

Weakness:

- Prior proteomic studies on the ECM of vertebrate organisms have shown the importance of allowing certain post-translational modifications during database search to ensure maximizing peptide-to-spectrum matching and accurately evaluating protein quantification. Such PTMs include the hydroxylation of lysines and prolines that are collagen-specific PTMs. Multiple reports have shown that omitting these PTMs while analyzing LC-MS/MS data would lead to underestimating the abundance of collagens and the misidentification of certain collagens. While the authors in their response state that the inclusion of these PTMs only led to a modest increase in protein identification, they do not comment on the impact of including these PTMs on PSMs or protein abundance (precursor ion intensity).

---

## [Author Response]

The following is the authors’ response to the original reviews.

**Reviewer #1 (Public review):**
Summary:In this manuscript entitled "Molecular dynamics of the matrisome across sea anemone life history", Bergheim and colleagues report the prediction, using an established sequence analysis pipeline, of the "matrisome" - that is, the compendium of genes encoding constituents of the extracellular matrix - of the starlet sea anemone *Nematostella vectensis*. Re-analysis of an existing scRNA-Seq dataset allowed the authors to identify the cell types expressing matrisome components and different developmental stages. Last, the authors apply time-resolved proteomics to provide experimental evidence of the presence of the extracellular matrix proteins at three different stages of the life cycle of the sea anemone (larva, primary polyp, adult) and show that different subsets of matrisome components are present in the ECM at different life stages with, for example, basement membrane components accompanying the transition from larva to primary polyp and elastic fiber components and matricellular proteins accompanying the transition from primary polyp to the adult stage.Strengths:The ECM is a structure that has evolved to support the emergence of multicellularity and different transitions that have accompanied the complexification of multicellular organisms. Understanding the molecular makeup of structures that are conserved throughout evolution is thus of paramount importance.The in-silico predicted matrisome of the sea anemone has the potential to become an essential resource for the scientific community to support big data annotation efforts and understand better the evolution of the matrisome and of ECM proteins, an important endeavor to better understand structure/function relationships. This study is also an excellent example of how integrating datasets generated using different -omic modalities can shed light on various aspects of ECM metabolism, from identifying the cell types of origins of matrisome components using scRNA-Seq to studying ECM dynamics using proteomics.

We greatly appreciate the positive feedback regarding the design of our study and the evolutionary significance of our findings.

Weaknesses:My concerns pertain to the three following areas of the manuscript:(1) In-silico definition of the anemone matrisome using sequence analysis:a) While a similar computational pipeline has been applied to predict the matrisome of several model organisms, the authors fail to provide a comprehensive definition of the anemone matrisome: In the text, the authors state the anemone matrisome is composed of "551 proteins, constituting approximately 3% of its proteome (see page 6, line 14), but Figure 1 lists 829 entries as part of the "curated" matrisome, Supplementary Table S1 lists the same 829 entries and the authors state that "Here, we identified 829 ECM proteins that comprise the matrisome of the sea anemone *Nematostella vectensis*" (see page 17, line 10). Is the sea anemone matrisome composed of 551 or 829 genes? If we refer to the text, the additional 278 entries should not be considered as part of the matrisome, but what is confusing is that some are listed as glycoproteins and the "new_manual_annotation" proposed by the authors and that refer to the protein domains found in these additional proteins suggest that in fact, some could or should be classified as matrisome proteins. For example, shouldn't the two lectins encoded by NV2.3951 and NV2.3157 be classified as matrisome-affiliated proteins? Based on what has been done for other model organisms, receptors have typically been excluded from the "matrisome" but included as part of the "adhesome" for consistency with previously published matrisome; the reviewer is left wondering whether the components classified as "Other" / "Receptor" should not be excluded from the matrisome and moved to a separate "adhesome" list.In addition to receptors, the authors identify nearly 70 glycoproteins classified as "Other". Here, does other mean "non-matrisome" or "another matrisome division" that is not core or associated? If the latter, could the authors try to propose a unifying term for these proteins? Unfortunately, since the authors do not provide the reasons for excluding these entries from the bona fide matrisome (list of excluding domains present, localization data), the reader is left wondering how to treat these entries.Overall, the study would gain in strength if the authors could be more definitive and, if needed, even propose novel additional matrisome annotations to include the components for now listed as "Other" (as was done, for example, for the *Drosophila* or *C. elegans* matrisomes).

The reviewer is correct to point out the confusing terminology used throughout our manuscript, where both the total of 829 proteins constituting the curated list of ECM domain proteins and the actual matrisome (excluding "others") were referred to as "matrisomes". In general, we followed the example set by Naba & Hynes in their 2012 paper (Mol Cell Proteomics. 2012 Apr;11(4):M111.014647. doi: 10.1074/mcp.M111.014647), where they define the "matrisome" as encompassing all components of the extracellular matrix ("core matrisome") and those associated with it ("matrisome-associated" proteins). This corresponds to our group of 551 proteins, comprising both core matrisome and matrisomeassociated proteins. The Naba & Hynes paper also contains the inclusive and exclusive domain lists for the matrisome that we applied for our dataset. In the revised manuscript, we have now labelled the group of 829 proteins as "curated ECM domain proteins/genes", which includes all proteins positively selected for containing a bona fide ECM domain. After excluding non-matrisomal proteins such as receptors, we arrive at the 551 proteins that constitute the "Nematostella matrisome". We have maintained this terminology throughout the revised manuscript and have revised Figures 1B and 4B accordingly.

Regarding the category of "other" proteins, which by definition are not part of the matrisome although containing ECM domains, we have taken the reviewer's advice and classified these in more detail. We categorized all receptors as "adhesome" (202 proteins). The remaining group of “other” secreted ECM domain proteins were then further subcategorized. Those exhibiting significant matches in the ToxProt database were subclassified as "putative venoms" (15 proteins). This group also includes the two lectins (NV2.3951 and NV2.3157), which had been originally shifted to the “other” category due to their classification as venoms. We categorized as “adhesive proteins” (28 proteins) factors such as coadhesins that due to their domain architecture resemble bioadhesive proteins described in proteomic studies of other invertebrate species, such as corals or sponges (see also https://doi.org/10.1016/j.jprot.2022.104506). Further sub-categories are stress/injury response proteins (9 proteins) and ion channels (6 proteins). The remaining 17 proteins were categorized as “uncharacterized ECM domain proteins”. These include highly diverse proteins possessing either single ECM domains or novel domain combinations. We decided to retain those in our dataset as candidates for future functional characterization.

b) It is surprising that the authors are not providing the full currently accepted protein names to the entries listed in Supplementary Table S1 and have used instead "new_manual_annotation" that resembles formal protein names. This liberty is misleading. In fact, the "new_manual_annotation" seems biased toward describing the reason the proteins were positively screened for through sequence analysis, but many are misleading because there is, in fact, more known about them, including evidence that they are not ECM proteins. The authors should at least provide the current protein names in addition to their "new_manual_annotations".c) To truly serve as a resource, the Table should provide links to each gene entry in the Stowers Institute for Medical Research genome database used and some sort of versioning (this could be added to columns A, B, or D). Such enhancements would facilitate the assessment of the rigor of the list beyond the manual QC of just a few entries.d) Since UniProt is the reference protein knowledge database, providing the UniProt IDs associated with the predicted matrisome entries would also be helpful, giving easy access to information on protein domains, protein structures, orthology information, etc.e) In conclusion, at present, the study only provides a preliminary draft that should be more rigorously curated and enriched with more comprehensive and authoritative annotations if the authors aspire the list to become the reference anemone matrisome and serve the community.

Table S1 has been updated to include links to the respective Stowers Institute IDs (first two columns), as well as SwissProt IDs and current descriptions from both the Stowers Institute (SI) and Swissprot.

In our manual annotations, we prioritized these over automated ones due to the considerable effort invested in examining each sequence individually. The cnidaria-specific minicollagens and NOWA proteins might serve as an example. According to the SI descriptions, the minicollagens are annotated as “keratin-associated protein, predicted or hypothetical protein, collagen-like protein and pericardin”. We classified these as minicollagens on the basis of overall domain architecture and of signature domains and sequence motifs, such as minicollagen cysteine-rich domains (CRDs) and polyproline stretches (doi: 10.1016/j.tig.2008.07.001). NOWA is a CTLD/CRD-containing protein that is part of nematocyst tubules (doi:10.1016/j.isci.2023.106291). The first two NOWA isoforms, according to Si descriptions, were annotated as aggrecan and brevican core proteins, which is very misleading. We therefore feel that our manual annotations better serve the cnidarian research community in classifying these proteins.

Automated annotations of ECM proteins often rely on similarities between individual domains, neglecting overall domain composition. For example, Swissprot descriptions annotate 31 TSP1 domain-containing proteins in our list as "Hemicentin-1", but closer inspection reveals that only one sequence (NV2.24790) qualifies as Hemicentin-1 due to its characteristic vWFA, Ig-like, TSP1, G2 nidogen, and EGF-like domain architecture. Regarding novel protein annotations, NV2.650 might serve as an example. While SI descriptions annotate this protein as "epidermal growth factor" based on the presence of several EGF-like domains, our analysis reveals two integrin alpha N-terminal domains that classify this sequence as integrin-related. We have therefore assigned a description (Secreted integrin-N-related protein) that references this defining domain and avoids misclassification within the EGF family.

In cases where the automated annotation (including those in Genbank) matched our own findings, we adopted the existing description, as seen with netrin-1 (NV2.7734). We acknowledge that our manual annotations are not flawless and will be refined by future research. Nonetheless, we offer them as an approximation to a more accurate definition of the identified protein list.

(2) Proteomic analysis of the composition of the mesoglea during the sea anemone life cycle:a) The product of 287 of the 829 genes proposed to encode matrisome components was detected by proteomics. What about the other ~550 matrisome genes? When and where are they expressed? The wording employed by the authors (see line 11, page 13) implies that only these 287 components are "validated" matrisome components. Is that to say that the other ~550 predicted genes do not encode components of the ECM? This should be discussed.

Obviously, our wording was not sufficiently accurate here. In the revised Fig. 1B we indicated that 210 of the 551 matrisome (core and associated) proteins were confirmed by mass spectrometry. In total, 287 proteins were identified by mass spectrometry, meaning that 77 of those are non-matrisomal proteins belonging to the “adhesome” (47) and “other” (30) groups. The fact that the remaining 542 proteins of the matrisome predicted by our in silico analysis could not be identified has two major reasons: (1) Our study was focussed on the molecular dynamics of the mesoglea. Therefore, only mesogleas were isolated for the mass spectrometry analysis and nematocysts were mostly excluded by extensive washing steps. As nematocysts contribute significantly to the predicted matrisome, this group of proteins is underrepresented in the mass spectrometry analysis. (2) A significant fraction of the predicted ECM proteins constitutes soluble factors and transmembrane receptors. These might not be necessarily part of the mesoglea isolates. In addition, the isolation and solubilization method we applied might have technical limitations. Although we used harsh conditions for solubilizing the mesoglea samples (90°C and high DTT concentrations), we cannot exclude that we missed proteins which resisted solubilization and thus trypsinization. We confirmed that all genes predicted by the in silico analysis have transcriptomic profiles as demonstrated in supplementary table S4. We have clarified these points in the revised results part (p.6) and also revised the statement in line 16, page 13.

b) Can the authors comment on how they have treated zero TMT values or proteins for which a TMT ratio could not be calculated because unique to one life stage, for example?

We did not include these proteins in the analysis of the respective statistical comparison. This involved only very few proteins (about 10).

c) Could the authors provide a plot showing the distribution of protein abundances for each matrisome category in the main figure 4? In mammals, the bulk of the ECM is composed of collagens, followed by fibrillar ECM glycoproteins, the other matrisome components being more minor. Is a similar distribution observed in the sea anemone mesoglea?

We have included such a plot showing protein abundances across life stages and protein categories (Fig. 4A). Collagens and basement membrane proteoglycans (perlecan) are the most abundant protein categories in the core matrisome while secreted factors dominate in the matrisome-associated group.

d) Prior proteomic studies on the ECM of vertebrate organisms have shown the importance of allowing certain post-translational modifications during database search to ensure maximizing peptide-to-spectrum matching. Such PTMs include the hydroxylation of lysines and prolines that are collagen-specific PTMs. Multiple reports have shown that omitting these PTMs while analyzing LC-MS/MS data would lead to underestimating the abundance of collagens and the misidentification of certain collagens. The authors may want to reanalyze their dataset and include these PTMs as part of their search criteria to ensure capturing all collagen-derived peptides.

Thank you for this suggestion. We have re-analyzed our dataset including lysine and proline hydroxylation as PTM. While we obtained in total 70 more proteins using this approach, this additional group did not contain any large collagen or minicollagen we had not detected before. We only obtained two additional collagen-like proteins with very short triple helical domains (V2t013973001.1, NV2t024002001.1), one being a fragment. We don’t feel this justifies implementing a re-analysis of the proteome in our study.

e) The authors should ensure that reviewers are provided with access to the private PRIDE repository so the data deposited can also be evaluated. They should also ensure that sufficient meta-data is provided using the SRDF format to allow the re-use of their LCMS/MS datasets.

We apologize for not providing the reviewer access in our initial submission and have asked the editorial office to forward the PRIDE repository link to all reviewers immediately after receiving the reviews. We did upload a metadata.csv file with the proteomics dataset. This file contains an annotation of all TMT labels to the samples and conditions and replicates used in the manuscript. It contains similar information as an SRDF format file. In addition, the search output files on protein and psm level have been provided. So, from our point of view, we provided all necessary information to reproduce the analysis.

(3) Supplementary tables:The supplementary tables are very difficult to navigate. They would become more accessible to readers and non-specialists if they were accompanied by brief legends or "README" tabs and if the headers were more detailed (see, for example, Table S2, what does "ctrl.ratio_Larvae_rep2" exactly refer to? Or Table S6 whose column headers using extensive abbreviations are quite obscure). Similarly, what do columns K to BX in Supplementary Table S1 correspond to? Without more substantial explanations, readers have no way of assessing these data points.

We have revised the tables and removed any redundant data columns. We also included detailed explanations of the used abbreviations, both in the headers and in a separate README file. Some of the information was apparently lost during the conversion to pdf files. We will therefore upload the original .xls files when submitting the revised manuscript.

**Reviewer #2 (Public review):**
This work set out to identify all extracellular matrix proteins and associated factors present within the starlet sea anemone *Nematostella vectensis* at different life stages. Combining existing genomic and transcriptomic datasets, alongside new mass spectometry data, the authors provide a comprehensive description of the Nematostella matrisome. In addition, immunohistochemistry and electron microscopy were used to image whole mount and decellularized mesoglea from all life stages. This served to validate the de-cellularization methods used for proteomic analyses, but also resulted in a very nice description of mesoglea structure at different life stages. A previously published developmental cell type atlas was used to identify the cell type specificity of the matrisome, indicating that the core matrisome is predominantly expressed in the gastrodermis, as well as cnidocytes. The analyses performed were rigorous and the results were clear, supporting the conclusions made by the authors.

Thank you. We greatly appreciate the positive assessment of our study.

**Reviewer #3 (Public review):**
Summary:This manuscript by Bergheim et al investigates the molecular and developmental dynamics of the matrisome, a set of gene products that comprise the extracellular matrix, in the sea anemone *Nematostella vectensis* using transcriptomic and proteomic approaches. Previous work has examined the matrisome of the hydra, a medusozoan, but this is the first study to characterize the matrisome in an anthozoan. The major finding of this work is a description of the components of the matrisome in Nematostella, which turns out to be more complex than that previously observed in hydra. The authors also describe the remodeling of the extracellular matrix that occurs in the transition from larva to primary polyp, and from primary polyp to adult. The authors interpret these data to support previously proposed (Steinmetz et al. 2017) homology between the cnidarian endoderm with the bilaterian mesoderm.Strengths:The data described in this work are robust, combining both transcriptome and proteomic interrogation of key stages in the life history of Nematostella, and are of value to the community.

Thank you for your positive assessment of our dataset.

Weaknesses:The authors offer numerous evolutionary interpretations of their results that I believe are unfounded. The main problem with extending these results, together with previous results from hydra, into an evolutionary synthesis that aims to reconstruct the matrisome of the ancestral cnidarian is that we are considering data from only two species. I agree with the authors' depiction of hydra as "derived" relative to other medusozoans and see it as potentially misleading to consider the hydra matrisome as an exemplar for the medusozoan matrisome. Given the organismal and morphological diversity of the phylum, a more thorough comparative study that compares matrisome components across a selection of anthozoan and medusozoan species using formal comparative methods to examine hypotheses is required.Specifically, I question the author's interpretation of the evolutionary events depicted in this statement:"The observation that in Hydra both germ layers contribute to the synthesis of core matrisome proteins (Epp et al. 1986; Zhang et al. 2007) might be related to a secondary loss of the anthozoan-specific mesenteries, which represent extensions of the mesoglea into the body cavity sandwiched by two endodermal layers."Anthozoans and medusozoans are evolutionary sisters. Therefore, the secondary loss of "anthozoan-like mesenteries" in hydrozoans is at least as likely as the gain of this character state in anthozoans. By extension, there is no reason to prefer the hypothesis that the state observed in Nematostella, where gastroderm is responsible for the synthesis of the core matrisome components, is the ancestral state of the phylum. Moreover, the fossil evidence provided in support of this hypothesis (Ou et al. 2022) is not relevant here because the material described in that work is of a crown group anthozoan, which diversified well after the origin of Anthozoa. The phylogenetic structure of Cnidaria has been extensively studied using phylogenomic approaches and is generally well supported (Kayal et al. 2018; DeBiasse et al. 2024). Based on these analyses, anthozoans are not on a "basal" branch, as the authors suggest. The structure of cnidarian phylogeny bifurcates with Anthozoa forming one clade and Medusozoa forming the other. From the data reported by Bergheim and coworkers, it is not possible to infer the evolutionary events that gave rise to the different matrisome states observed in Nematostella (an anthozoan) and hydra (a medusozoan). Furthermore, I take the observation in Fig 5 that anthozoan matrisomes generally exhibit a higher complexity than other cnidarian species to be more supportive of a lineage-specific expansion of matrisome components in the Anthozoa, rather than those components being representative of an ancestral state for Cnidaria. Whatever the implication, I take strong issue with the statement that "the acquisition of complex life cycles in medusozoa, that are distinguished by the pelagic medusa stage, led to a secondary reduction in the matrisome repertoire." There is no causal link in any of the data or analyses reported by Bergheim and co-workers to support this statement and, as stated above, while we are dealing with limited data, insufficient to address this question, it seems more likely to me that the matrisome expanded in anthozoans, contrasting with the authors' conclusions. While the discussion raises many interesting evolutionary hypotheses related to the origin of the cnidarian matrisome, which is of vital interest if we are to understand the origin of the bilaterian matrisome, a more thorough comparative analysis, inclusive of a much greater cnidarian species diversity, is required if we are to evaluate these hypotheses.DeBiasse MB, Buckenmeyer A, Macrander J, Babonis LS, Bentlage B, Cartwright P, Prada C, Reitzel AM, Stampar SN, Collins A, et al. 2024. A Cnidarian Phylogenomic Tree Fitted With Hundreds of 18S Leaves. Bulletin of the Society of Systematic Biologists [Internet] 3. Available from: https://ssbbulletin.org/index.php/bssb/article/view/9267Epp L, Smid I, Tardent P. 1986. Synthesis of the mesoglea by ectoderm and endoderm in reassembled hydra. J Morphol [Internet] 189:271-279. Available from: https://pubmed.ncbi.nlm.nih.gov/29954165/Kayal E, Bentlage B, Sabrina Pankey M, Ohdera AH, Medina M, Plachetzki DC, Collins AG, Ryan JF. 2018. Phylogenomics provides a robust topology of the major cnidarian lineages and insights on the origins of key organismal traits. BMC Evol Biol [Internet] 18:1-18. Available from: https://bmcecolevol.biomedcentral.com/articles/10.1186/s12862-018-1142-0Ou Q, Shu D, Zhang Z, Han J, Van Iten H, Cheng M, Sun J, Yao X, Wang R, Mayer G. 2022. Dawn of complex animal food webs: A new predatory anthozoan (Cnidaria) from Cambrian. The Innovation 3:100195Steinmetz PRH, Aman A, Kraus JEM, Technau U. 2017. Gut-like ectodermal tissue in a sea anemone challenges germ layer homology. Nature Ecology & Evolution 2017 1:10 [Internet] 1:1535-1542. Available from: https://www.nature.com/articles/s41559-017-0285-5Zhang X, Boot-Handford RP, Huxley-Jones J, Forse LN, Mould AP, Robertson DL, Li L, Athiyal M, Sarras MP. 2007. The collagens of hydra provide insight into the evolution of metazoan extracellular matrices. J Biol Chem [Internet] 282:6792-6802. Available from: https://pubmed.ncbi.nlm.nih.gov/17204477/

We agree with the reviewer that only the analysis of several additional anthozoan and medusozoan representatives will yield a valid basis for a reconstruction of the ancestral cnidarian matrisome and allow statements about ancestral or novel features within the phylum. We have therefore revised our statements in the discussion part of the manuscript by implementing the cited literature and also findings from medusozoan genome analysis (e.g. Gold et al., 2018) demonstrating that changes in gene content are as common in the anthozoans as in medusozoans, which questioned the previously stated “basal” state of Nematostella or of anthozoans in general.

**Reviewer #1 (Recommendations for the authors):**
(1) In Figure 2A, an "o" is missing in the labeling of the "developing cnidcytes" population.

Thank you, we have corrected the typo.

(2) It would be helpful to have the different life stages indicated as headers of the heat maps presented in Figure 4.

We have included symbolic representations for the different life stages on top of the heat maps in addition to the respective labels at the bottom.

**Reviewer #2 (Recommendations for the authors):**
Important changes:(1) Figure 2B The x-axis tissue names should be changed to something more easily readable/understandable - some are clear, but others are not. Perhaps abbreviations could be expanded in the legend.

We have expanded the legend in Fig. 2B to render it more easily readable. We have also rotated the maps in A to have them aligned with the ones in Fig.3B.

(2) Figure 3B This figure would be improved by the inclusion of cluster names, to understand better the mapping.

We have added relevant cluster names to Fig. 3B and as stated above aligned the orientation of the maps in Fig. 2B and Fig. 3B.

(3) Figure 3C As with 2B, I find the y-axis cnidocyte cell state names to be unclear at times. Perhaps abbreviations could be expanded in the legend.

All abbreviations were expanded in Fig.3C axis labels.

(4) Many of the supplementary tables are not well exported or easily readable as is (gene names are truncated, headers truncated, etc), which means that they may not be easily usable by researchers in the field interested in following up on this work in other contexts. Indeed, to be more usable, please consider sharing these supplementary data as .csv files, for example, instead of as .pdfs.

We are sorry for this inconvenience, which was obviously caused by the conversion to pdf files. We will upload the original csv files when submitting the revised manuscript.

Smaller nitpicky comments:(5) Page 2 line 4 & page 3 line 7: Please consider a term other than "pre-bilaterian". The drawing/ordering of a phylogeny of extant species is not meaningful in terms of more or less ancestral. e.g. if the tips are flipped in the drawing of the tree, can we say that bilaterians are pre-cnidarians? What does that mean?

We have used that term on the basis that cnidarians existed before the appearance of bilaterians according to the fossil record and molecular phylogenies (McFadden et al., 2021; Adoutte et al., 2000;Cavalier-Smith et al., 1996; Collins, 1998; Kim et al., 1999; Medina et al., 2001; Wainright et al., 1993). To acknowledge remaining uncertainties in the timing of origin of animals, we will use the term “early-diverging metazoans” instead, which is widely accepted in the cnidarian community.

(6) Page 3 line 9 I was confused by the use of "gastrula-shaped body" to describe cnidarians, which are on the whole very morphologically diverse and don't all resemble gastrulae (that can also be quite diverse).

This term is sometimes used to refer to the diploblastic cnidarian body plan (outer ectoderm, inner endoderm) with a mouth that corresponds to the blastopore. To avoid misunderstandings, we changed it in the revised manuscript to “Cnidarians, the sister group to bilaterians, are characterized by a simple body plan with a central body cavity and a mouth opening surrounded by tentacles.”

**Reviewer #3 (Recommendations for the authors):**
(1) In general, I felt there was a lot of discussion about protein structure and diversity that is difficult to follow without a figure. I think some of the information in Supplementary Figures S5, S9, and S11 should be in the main figures.

Following the reviewer’s suggestion, we have integrated Fig. S5 (collagens) into the main Fig. 2 and Fig. S9 (polydoms) into Fig. 4. As metalloproteases are not extensively discussed in the manuscript (and also due to the large size of the figure) we have kept Fig. S11 as a supplementary figure.

(2) Page 3, Line 7: The use of the term "pre-bilaterian" is inappropriate. Cnidarians and bilaterians are evolutionary sisters. Therefore, each lineage derives from the same split and is the same age. The cnidarian lineage is not older than the bilaterian lineage.

Following a similar request by reviewer 2 we have replaced this term by “early diverging metazoans”.

(3) Page 5, Line 10. How were in silico matrisomes from early-branching metazoan species predicted?

We applied the same bioinformatic pipeline as for the Nematostella matrisome. We clarified this in the respective methods part.

(4) Page 16, Line 8: This should be Thus.

Obviously, the wording of this sentence was ambiguous. We changed it to ”In contrast, the adult mesoglea is significantly enriched in elastic fiber components, such as fibrillins and fibulin. This compositional shift likely adds to the visco-elastic properties (Gosline 1971a, b) of the growing body column (Fig. 4B,D, supplementary table S7).”